# EXECUTABLE NETWORKS OF THOUGHT: SCALING REASONING WITH LLM WORKFLOW TEMPLATE

## ABSTRACT

Past prompt schemes, such as Chain-of-Thought (CoT) and Tree of Thoughts (ToT), either lack modularity or rely on manually engineered, task-specific prompts and fixed solution structures, limiting their *scalability*. To overcome these limitations, we propose **Executable Network of Thoughts (XNoT)**, a prompt scheme that leverages LLMs' intrinsic capabilities to autonomously plan and execute reasoning steps from minimal user input. Central to XNoT is the **LLM Workflow Template (LWT)**, a format that supports a network of thought dependencies among sequential elementary steps, enabling XNoT to flexibly adapt to different task complexities and input lengths. XNoT demonstrates superior scalability compared to prior methods. For example, while all methods achieve near 100% accuracy on sorting 16 numbers, XNoT attains 92% on sorting 32 numbers, substantially outperforming CoT (0%), ToT (12%).

## 1 INTRODUCTION

Large language models (LLMs) demonstrates a level of reasoning (Qiao et al., 2023) but falter on multi-step reasoning unless given a clear plan or scaffold (Wei et al., 2022). They suffer from the hallucination problem, where they produce plausible-sounding yet incorrect output (Huang et al., 2025), and can be distracted by irrelevant parts of the input (Shi et al., 2023). To improve multi-step reasoning, prompt engineering (Definition 1) builds *prompt schemes* that pair fixed instruction text with an explicit algorithmic procedure (Liu et al., 2023). These schemes define a small, principled set of actions for the model to execute over several intermediate steps—often called *thoughts*, i.e., discrete units in a multi-step solution (Wei et al., 2022; Besta et al., 2024a;b). Consequently, recent studies concentrate on mathematical and logical reasoning tasks (Zhou et al., 2023; Shin and Kim, 2025). Success in these settings shows that, under well-designed prompts, LLMs can plan, execute, and check intermediate steps rather than jumping to a final guess, leading to more reliable solutions.

However, previous works often overlook the *scalability* issue in prompt engineering. As explained in Definition 2, scalability concerns whether a prompt scheme remains effective as the input size increases (Chen et al., 2025).[1] Existing methods generally fall into two categories. Monolithic prompting methods—e.g., Chain of Thought (CoT) (Wei et al., 2022) and Least to Most (L2M) (Zhou et al., 2023)—perform a single (Wei et al., 2022; Kojima et al., 2022; Wang et al., 2023) or fix number (Wang et al., 2022; Zhou et al., 2024; Dua et al., 2022; Zhou et al., 2023) of inference passes. Such designs usually scale poorly, as they lack modularity, i.e., the separation of reasoning steps into distinct inference passes, and are limited to chain-like structures that force information to accumulate in later steps, increasing the risk of error propagation (Besta et al., 2024b). Algorithmic prompting methods—e.g., Tree of Thoughts (ToT) (Yao et al., 2024) and Graph of Thoughts (GoT) (Besta et al., 2024a)—in contrast, organize reasoning as trees or graphs, coordinating multiple operations through external control scripts, i.e., Python modules. While these methods improve modularity by separating reasoning steps into separate LLM inferences, they also introduce and rely on rigid, hard-coded manual workflows that make it difficult to adapt to varying instance sizes. Consequently, both categories struggle to scale reliably as task size increase.

---

[1] While recent works aim to increase the context length (Chen et al., 2023a; 2024; Ding et al., 2024) to improve LLM performances over long inputs (Levy et al., 2024), a larger context length does not inherently guarantee effective multi-step reasoning (Liu et al., 2024).

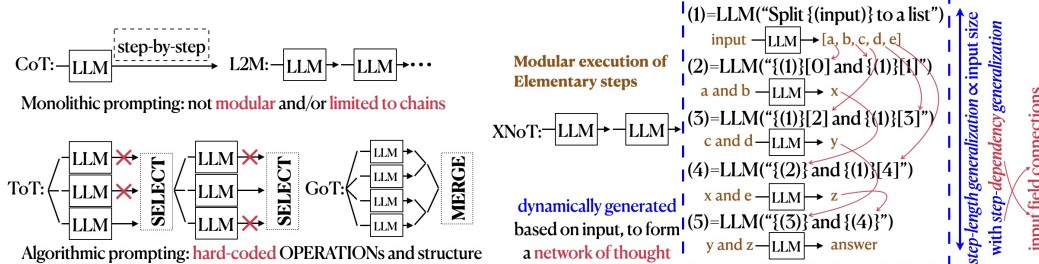

Figure 1: Prompt scheme comparison. While some monolithic prompting methods introduce modular execution (e.g., L2M), they remain limited to linear, chain-like structures. Algorithmic prompting supports networked reasoning but relies on rigid, hard-coded scripts. In contrast, XNoT dynamically generates a *network of thought* without requiring predefined workflows or complex control scripts.

To address the scalability issue, it is important for a prompt scheme to demonstrate two key capabilities: 1) *step-length generalization*—the ability to scale the number of reasoning steps while keeping each step *elementary*, i.e., the minimal reasoning units into which the problem can be decomposed, to effectively reduce error (Zhou et al., 2023; Xue et al., 2024) (see the theoretical analysis in Section 5); and 2) *step-dependency generalization*—the ability to maintain correct dependencies between steps and coordinate them through network-structured reasoning, correctly recombining the decomposed steps into a final answer as the instance size increases (Zhou et al., 2024; Yao et al., 2024; Besta et al., 2024a). Real-world tasks vary in size and complexity, requiring scalable prompt schemes (Ji et al., 2023). For example, in code refactoring, a programmer may efficiently update a small script by modifying multiple components at once. However, for a larger codebase, breaking it into steps that process one component at a time reduces errors, but the coordination complexity between components usually grows beyond linear chains or trees as the system scales.

In this paper, we introduce *Executable Network of Thoughts (XNoT)*, a prompt scheme that addresses the scalability issue in LLM reasoning through *intelligence amplification*. Our idea is to transform prompt engineering from manually hard-coding complete workflows to providing the minimal scaffolding that LLMs can autonomously expand. While approaches like ToT (Yao et al., 2024) and GoT (Besta et al., 2024a) require humans to predefine rigid structures for each task size, XNoT shifts this burden to LLMs by enabling it to infer both task decomposition and step coordination from simple examples. This allows step-length generalization, as LLM dynamically determines an appropriate number of elementary steps without requiring humans to redesign scripts for different problem sizes. As input complexity increases, XNoT enables LLMs to naturally expand its reasoning network with additional elementary steps to match the growing demands of the task.

At the core of XNoT is the *LLM Workflow Template (LWT)*, a structured text format that enables step-dependency generalization through a simple notation system. LWT allows each reasoning step to reference outputs from any previous step, forming flexible networks of thought with precisely defined dependencies. Unlike the rigid chains (Wei et al., 2022), trees (Yao et al., 2024), or split-then-merge graphs (Besta et al., 2024a) used in prior work, LWT empowers XNoT to construct instance-specific reasoning networks that capture the exact dependencies required for each query. By combining elementary step decomposition with dynamic dependency management, XNoT achieves *intelligence amplification*: LLM generalizes the user-specified minimal examples to autonomously build comprehensive solution structures for previously unseen problems, eliminating the need for hard-coded workflows of each task variation.

Experiments demonstrate that XNoT scales much more reliably to larger input sizes. For example, when the sorting task scales from 16 to 64 numbers, most baselines drop to 0% accuracy, whereas XNoT maintains 27%. XNoT also significantly reduces manual prompt engineering effort (up to 84.4% less than ToT and 87.3% less than GoT) and lowers the API costs compared to sophisticated algorithmic prompting methods. In addition, we provide theoretical analyses showing how decomposition into elementary steps improves performance under super-linear accuracy decay, while dynamic merge strategies reduce computational overhead compared to fixed structures.

Table 1: Comparing full (■), partial (◧), or no support (✗) for each prompt scheme capacity.

| Prompt Schemes | Modularized Steps? | Elementary Steps? | Network of Thought? | Intelligence Amplification? |
|---|---|---|---|---|
| Chain of Thoughts (CoT) Wei et al. (2022) | ✗ | ✗ | ✗ | ✗ |
| Zero-shot Chain of Thoughts (Zero-CoT) Kojima et al. (2022) | ✗ | ✗ | ✗ | ✗ |
| Plan-and-solve (PS) Wang et al. (2023) | ✗ | ✗ | ✗ | ✗ |
| Self-consistency Chain of Thoughts (CoT-SC) Wang et al. (2022) | ✗ | ✗ | ✗ | ✗ |
| Automatic Chain of Thoughts (Auto-CoT) Zhang et al. (2023) | ✗ | ✗ | ✗ | ✗ |
| Self-discover (SD) Zhou et al. (2024) | ◧ | ✗ | ✗ | ✗ |
| Least to Most (L2M) Zhou et al. (2023) | ■ | ◧ | ✗ | ✗ |
| Successive Prompting (SP) Dua et al. (2022) | ■ | ◧ | ✗ | ✗ |
| Skeleton of Thoughts (SoT) Ning et al. (2023) | ■ | ◧ | ✗ | ✗ |
| Tree of Thoughts (ToT) Yao et al. (2024) | ■ | ◧ | ◧ | ✗ |
| Algorithm of Thought (AoT) Sel et al. (2024) | ■ | ◧ | ◧ | ✗ |
| Graph of Thoughts (GoT) Besta et al. (2024a) | ■ | ◧ | ◧ | ✗ |
| Tree of Problem (ToP) Zebaze et al. (2024) | ■ | ◧ | ◧ | ✗ |
| Enhancing Graph Of Thought (EGoT) Shin and Kim (2025) | ■ | ◧ | ◧ | ✗ |
| Executable Network of Thoughts (XNoT) | ■ | ■ | ■ | ■ |

## 2 RELATED WORK

To enable effective reasoning in LLMs, researchers have proposed *monolithic prompting*, which arranges steps in a simple sequence. Chain-of-Thought (CoT) (Wei et al., 2022) pioneers this approach by guiding models to produce intermediate steps in a single pass, instructing LLMs with "Let's think step-by-step" and providing few-shot examples. Later work shows even a single instruction (e.g., "Let's think step-by-step" (Kojima et al., 2022), or more structured plan-then-solve instructions (Wang et al., 2023)) can boost performance without few-shot examples. Self-Consistent CoT (CoT-SC) (Wang et al., 2022) improves reliability by aggregating multiple runs, while Auto-CoT (Zhang et al., 2023) automates few-shot example generation for CoT using Zero-CoT.

However, these methods remain constrained because they do not leverage *modularized steps*: the multiple intermediate steps are generated within a single-pass, with no explicit control, intermediate result reuse, or branching. Such approach limits the construction of flexible reasoning *networks* that coordinate results across passes. Subsequent methods, such as Least-to-Most (L2M) (Zhou et al., 2023), Successive Prompting (SP) (Dua et al., 2022), Self-Discover (SD) (Zhou et al., 2024), and Skeleton-of-Thoughts (SoT) (Ning et al., 2023), increase modularity by decomposing tasks across passes, but still enforce linear pipelines and lack mechanisms for coordinating multi-branch reasoning or handling complex dependencies.

On the other hand, *algorithmic prompting* methods rely on complex control scripts to manage more elaborate reasoning structures. Tree of Thoughts (ToT) (Yao et al., 2024) introduces tree-structured reasoning following the breadth-first search algorithm, while Algorithm of Thought (AoT) (Sel et al., 2024) extends it with the depth-first search algorithm. Both require external evaluators to score and select branches at each step, adding multiple LLM calls and increasing computational cost. Graph of Thoughts (GoT) (Besta et al., 2024a) generalizes trees to graphs via split-merge pipelines, while Tree of Problems (ToP) (Zebaze et al., 2024) simplifies GoT by removing explicit scoring, and Enhancing Graph of Thought (EGoT) (Shin and Kim, 2025) augments GoT with rationale propagation. Despite these advances, such methods remain tied to rigid, task-specific algorithms and hard-coded scripts, limiting flexibility and scalability.[2]

In contrast, we introduce a lightweight control framework for XNoT that supports richer networked reasoning than algorithmic prompting by leveraging intelligence amplification to generate LWT-formatted reasoning plans (Fig.1). Beyond supporting *modular* and *elementary steps* within a *network of thoughts*, XNoT enables LLMs to construct complete solution plans from their own knowledge, given only a single one-shot LWT example. This greatly reduces the need for manual task-specific redesign, allowing LLMs to *scale* to larger query inputs of the same task without requiring length-specific scripts. Table 1 summarizes the comparison, while Appendix A provides a more detailed and extended discussion. In particular, Appendix A.5 contrasts XNoT with multi-agent systems (MAS), highlighting its prompt-native focus versus MAS's role-based orchestration.

---

[2]Specifically, ToT, AoT, GoT, ToP require both few-shot examples and hand-crafted scripts; EGoT drops the former by rationale propagation, but still requires the latter. In contrast, XNoT eliminates both, using a one-shot LWT example and LLM-generated LWT script as the algorithm, achieving *intelligence amplification*.

## 3 PROBLEM FORMULATION

In this work, we aim to reduce human effort in prompt engineering by applying *intelligence amplification*, enabling LLMs to assume a larger share of the workload in complex reasoning tasks. To this end, we first provide a formal characterization of the human effort required in prompt engineering.

**Definition 1** (Prompt Engineering of LLM). *Let $LLM$ be a pretrained LLM and $\mathcal{T} = \{t_1, \ldots, t_n\}$ a set of reasoning tasks. For each task $t \in \mathcal{T}$, let $\mathcal{D}_t = \{(\mathbf{q}_i^t, \mathbf{a}_i^t)\}_{i=1}^{N_t}$ be the evaluation set, where $\mathbf{q}_i^t$ is the input query, $\mathbf{a}_i^t$ the corresponding answer, and $N_t$ the number of samples for task $t$.*

*The Prompt Engineering Problem seeks to design a prompt scheme consisting of an algorithm $\mathcal{A}$ that sequences LLM calls, a collection of constant prompts $\mathcal{P}_{const}$ reusable across tasks, and a collection of task-specific prompts $\mathcal{P}_{task}^t$ for each $t \in \mathcal{T}$, such that*

$$\mathcal{A}(LLM, \mathcal{P}_{const}, \mathcal{P}_{task}^t, \mathbf{q}_i^t) = \mathbf{a}_i^t, \ \forall t \in \mathcal{T}, i \in [1, N_t], \tag{1}$$

*where $\mathcal{A}$ may invoke the LLM multiple times, integrating $\mathcal{P}_{const}$, $\mathcal{P}_{task}^t$, $\mathbf{q}_i^t$, and intermediate results.*

Prompt scheme development involves three essential phases. **1) Prompt scheme design** specifies the algorithm $\mathcal{A}$ and constant prompts $\mathcal{P}_{const}$, establishing the level of modularization and potential for networked reasoning. **2) Task-specific design** develops task-specific prompts $\mathcal{P}_{task}^t$ for each task $t$, typically including task instructions and few-shot examples. **3) Execution** applies the prompt scheme by running the algorithm $\mathcal{A}$ together with $\mathcal{P}_{const}$ and $\mathcal{P}_{task}^t$ to generate task answers.

Developing a prompt scheme requires substantial human efforts, particularly in phase 2, which usually becomes the bottleneck when adapting to new tasks. This challenge worsens when tasks include instances of varying sizes, such as arithmetic sequences of length 8 or 32, referred to as different *divisions* of the same task. Algorithmic prompting methods are especially rigid, requiring separate Python scripts and task-specific prompts for each division, limiting flexibility and scalability. In contrast, XNoT enables the LLM to dynamically generate part of the algorithm $\mathcal{A}$ as a LWT script, reducing the need for hard-coded procedures. To address the above, we formally define the scalability problem, focusing on generalization across divisions with minimal human intervention.[3]

**Definition 2** (Scalability of Prompt Engineering). *Given different divisions of the same task, the scalability problem concerns the ability of a prompt scheme to generalize effectively across these divisions while sustaining performance as instance sizes increase. This involves two critical aspects: i) minimizing human effort required to adapt the prompt scheme across divisions, and ii) maintaining reliable performance on larger instance sizes.*

To concretize this notion of scalability, we evaluate across benchmark tasks that embody distinct reasoning challenges: **natural language task** (e.g., keyword counting) benefits from decomposing the text into smaller units, but requires careful aggregation of partial results to ensure correct global counting; **symbolic tasks** (e.g., sorting with duplicates, set intersection) demand efficient intermediate storage and accurate element-wise examination, which become increasingly complex as input size grows; **numerical tasks** (e.g., arithmetic calculations, large-digit addition) involve intricate carry-over management and digit-level operations that scale poorly with longer sequences. [4]

## 4 EXECUTABLE NETWORK OF THOUGHTS

Executable Network of Thoughts (XNoT) aims to (i) reduce manual prompt design via *intelligence amplification*, (ii) support flexible *networks of thought*, and (iii) operate through *elementary steps* for precise execution. Concretely, XNoT leverages LLMs to facilitate and automate the production of a task-specific *solution plan* and compile it into a *LWT script*. The LWT script is then used as the final prompt scheme design that is executed sequentially in separate inference passes to obtain the final answer. The result is modularization without rigid, hand-written control code: plans are authored by the LLM and realized as an executable prompt-native program.

To streamline the execution of each reasoning step, XNoT employs the LLM Workflow Template (LWT), a specialized prompt format that defines placeholders (*input fields*) for referencing outputs from previous steps, supports selecting items in list-based outputs (*indexing*), and specifies

---

[3]Additional discussions on how XNoT alleviates human labor are provided in Appendix A.3.

[4]We further examine *Game of 24*, extending it from 4-numbers to 5 numbers, in Appendix D.1.4.

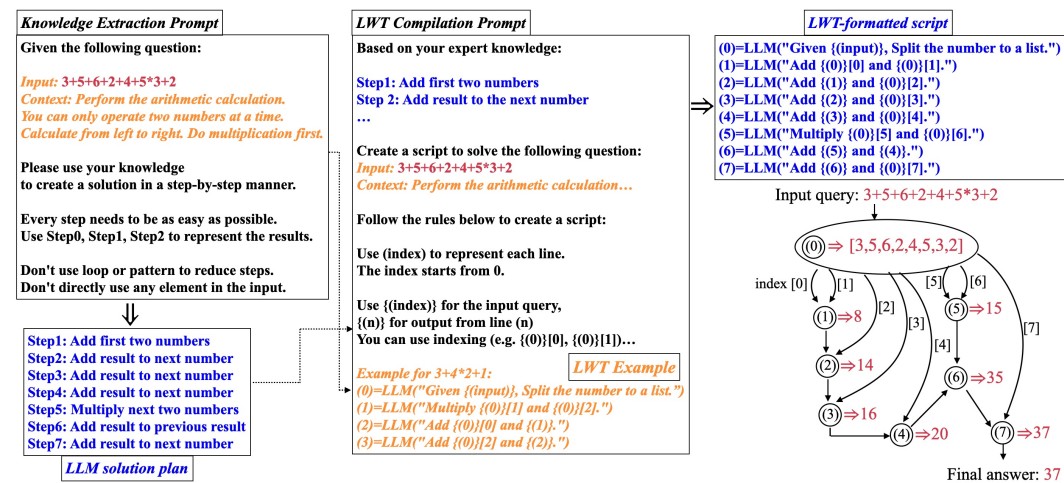

Figure 2: Illustration of Executable Network of Thoughts (XNoT). XNoT first generates an LLM solution plan based on the input query, then compiles it into an LWT-formatted script. The bottom-right diagram shows how the LWT-formatted script executes, forming a network of thought processes. Straight arrows indicate message passing through input fields, curved arrows indicate indexed field selection, and ⇒ marks LLM operations. Colors highlight different prompt types: constant prompts in black, task-specific prompts in orange, and LLM-generated prompts in blue.

the required operation (*instruction*). By prompting LLMs to devise fine-grained *elementary steps* in LWT-formatted scripts and orchestrate these steps into a generalized *knowledgeable network of thought*, XNoT enables flexible and robust problem-solving while avoiding the extensive manual engineering to build separate task-specific scripts for different instance sizes, required by prior prompt schemes (Sel et al., 2024; Yao et al., 2024; Zebaze et al., 2024; Besta et al., 2024a).

## 4.1 LLM WORKFLOW TEMPLATE (LWT)

LWT facilitates *networks of thought* by forming a *message-passing network* between LWT-instructions. Each instruction represents a *reasoning step*, executed in a separate *inference pass* for full *modularization*. LWT defines "input fields" that receive outputs from previous instructions, forming the directed edges of the network. To enable *elementary steps*, LWT provides indexing to select individual elements from the query, supporting operations at the element level.

An LWT-formatted script consists of a list of LWT-instructions, where each instruction may reference outputs from any preceding instructions using the following notation:

1. "(X)=LLM(...)" marks the $X^{th}$ LWT-instruction.

2. $\{(N)\}$ denotes an input field that receives the entire output of the $N^{th}$ LWT-instruction.

3. $\{(N)\}[I]$ selects the $I^{th}$ item when $\{(N)\}$ is a list.

An example of a LWT-instruction with X>N is illustrated below:

(X)=*LLM("Example LWT-instruction with input field* $\{(N)\}$ *and indexed input field* $\{(N)\}[I]")$

The proposed LWT format aligns with LLMs' inference behavior by structuring reasoning as a sequence of modular steps with explicit data dependencies. Equipped with references like $\{(N)\}$ and $\{(N)\}$, it defines a directed acyclic reasoning graph entirely in natural language, enabling both LLM generation and accurate parsing for step-by-step execution. This format encourages elementary operations, reduces ambiguity, and supports long-range connections, thereby improving stability and generalization. As a prompt-native abstraction, it can be integrated seamlessly into LLM workflows, supporting efficient, scalable, and interpretable execution across diverse reasoning tasks.

**Algorithm 1** Executable Network of Thoughts.

**Input:** query $\mathbf{q}$, context $C$, LWT example $\mathcal{E}$, knowledge extraction prompt $\mathcal{K}$, script compilation prompt $\mathcal{T}$.
**Procedure:**
1: $\mathfrak{P} \leftarrow LLM(\mathcal{K} \oplus \{\mathbf{q}, C\})$                                              //Extract LLM solution plan.
2: $\mathcal{S} \leftarrow LLM(\mathcal{T} \oplus \{\mathbf{q}, C, \mathcal{E}, \mathfrak{P}\})$               //Compile solution plan into LWT-formatted script.
3: Initialize list $L$ as empty                                      //Prepare for sequential script execution.
4: **for** each LWT-instruction in $\mathcal{S}$ **do**
5:     Resolve input fields by referencing $L[\text{x}][\text{I}]$ if indexed, else $L[\text{x}]$                         //Section 4.1.
6:     output $\leftarrow LLM(\text{LWT-instruction})$; append output to $L$               //Store intermediate result.
7: **return** output

## 4.2 Detailed XNoT Mechanism

Building on the LWT format, we detail the *Executable Network of Thoughts (XNoT)*. Fig. 2 illustrates XNoT applied to an example arithmetic task.[5] XNoT features a three-step process: *1) knowledge extraction*, which captures the LLM's knowledge as an *LLM solution plan* $\mathfrak{P}$; *2) LWT compilation*, which converts $\mathfrak{P}$ into an executable *LWT-formatted script* $\mathcal{S}$; and *3) script execution*, which sequentially performs the LWT-instructions in $\mathcal{S}$ to generate the final answer.

The LWT-formatted script $\mathcal{S}$ corresponds to the *task-instruction* component of the task-specific prompts $\mathcal{P}_{task}^{t}$ for a task $t$ (see Definition 1). Notably, XNoT automates the generation of $\mathcal{S}$ through its first two stages, substantially reducing the human effort typically required in **Task-specific design** and alleviating the bottleneck for scalable adaptation across divisions. Algorithm 1 presents the pseudo-code, and the prompt texts are provided in Appendix B.

**Stage I: Knowledge Extraction.** As shown on the left of Fig. 2, Stage I generates an LLM solution plan $\mathfrak{P}$ using a *Knowledge Extraction Prompt* $\mathcal{K}$, which integrates the input query $\mathbf{q}$, a task description $C$, and *extraction prompts*. The task description $C$ serves as $\mathcal{P}_{task}$ (Definition 1), defining the objective and providing hints to guide plan generation. For example, in arithmetic, the LLM is prompted to proceed step-by-step (e.g., two numbers at a time) and prioritize multiplication.

The *extraction prompts* are constant prompts $\mathcal{P}_{const}$, including a *node simplicity prompt* and an *edge simplicity prompt*. The node simplicity prompt promotes *elementary steps* by directing the LLM to produce each step "as easy as possible," without which the LLM might attempt to solve the entire problem in one step and thereby suffer from the lack of modularity (Levy et al., 2024). The *edge simplicity prompt* guides the LLM to identify the simplest valid dependencies between steps, avoiding unnecessary control structures and ensuring a clear, sequential operation order for LWT compilation. This modularization reduces error propagation often seen in long, unstructured outputs. Finally, the LLM is instructed to references query elements by position, reducing hallucination risk and aligning with LWT's indexing notation.

**Stage II: Script Compilation.** As shown in the middle of Fig.2, $\mathfrak{P}$ is compiled into a LWT-formatted script $\mathcal{S}$ using the *LWT Compilation Prompt* $\mathcal{T}$, incorporating $\mathfrak{P}$, *compilation instructions*, LWT example $\mathcal{E}$. The compilation instructions (which are constant prompts $\mathcal{P}_{const}$) specify the LWT format to guide the LLM in producing valid syntax. The LWT example $\mathcal{E}$ (which are task-specific $\mathcal{P}_{task}$) demonstrates valid syntax and dependency structure on a small instance, enabling generalization to the current (larger) input.[6] To facilitate elementary steps that parse query components, XNoT design the first LWT-instruction to break the query into its elements (e.g., listing values for sorting tasks), allowing subsequent instructions to index individual elements by referencing the output of the first step. Because XNoT automatically generates the LWT-formatted script from the LLM's own solution plan, it eliminates the need for manual scripting for each new task division. This fully automated generation of the LWT-formatted script stands in contrast to algorithmic prompting methods like ToT and GoT, which require manually defined control flows or Python scripts for each task or input size. By shifting this design burden to the LLM, XNoT reduces human effort and scales more easily across varying task divisions without manual redesign.

**Stage III: LWT Script Execution.** Finally, the LWT-formatted script $\mathcal{S}$ is executed sequentially. XNoT initializes a List $L$ as a cache to store intermediate outputs (Line 13 in Algorithm 1). Specif-

---

[5]Full prompt texts are provided in Appendix B.1.

[6]E.g., 4-number arithmetic case is provided for $16/32/64$ numbers.

ically, $L$ appends each node's output as the execution proceeds. If a LWT-instruction contains an input field $\{N\}$, the value is retrieved as $L[N]$. If the input field is indexed as $\{N\}[I]$, the value is retrieved as $L[N][I]$, since $L[N]$ holds a list. The LLM then executes the formatted LWT-instruction, and the output is appended to $L$ for later use. This mechanism enables later instructions to selectively reference earlier outputs, either fully or element-wise. Multiple input fields can appear in different positions within a single instruction, allowing flexible composition of the partial results. The output of the final instruction is returned as the answer to query $\mathbf{q}$. Unlike monolithic prompting, which processes all steps linearly and struggles to manage complex dependencies, XNoT executes a flexible network of thought, enabling more reliable reasoning on longer and more structured inputs.

## 5  THEORETICAL ANALYSIS

While LLM accuracy usually decays *super-linearly* with query length (Liu et al., 2024; Levy et al., 2024), XNoT mitigates this issue by (i) decomposing long queries into short, context-efficient sub-queries and (ii) merging partial results through a LWT index. We now formalize the conditions under which decomposition improves end-to-end success probability with the decay exponent $\gamma$. All proofs are presented in Appendix C.

**Theorem 1** (Decomposition Benefit). *LLM processes a query of length $L$ correctly with probability*

$$P(L) = \exp\!\big(-a\,L^{\gamma}\big),\ a, \gamma > 0. \tag{2}$$

*If we decompose a length-$L$ task into $K > 1$ equal parts of size $L/K$, then the joint probability of correctly executing the decomposed parts surpasses that of the full-sequence iff $\gamma > 1$.*

Theorem 1 isolates the *accuracy* benefit of decomposition: splitting pays off precisely when long-context decay is super-linear ($\gamma > 1$). To turn this potential benefit into *net* gain, we analyze the *cost* of the extra LLM calls induced from recombination.

**Lemma 1** (Recombination overhead). *Suppose a reasoning task of length $L$ is divided into $K$ segments, each of length $L/K$. Then, the total number of LLM calls to solve the task may be proportional $K^p$ for some overhead exponent $p \geq 0$ due to the need to consolidate each intermediate steps into the final answer. For instance, a pairwise merging over the $K$ segments (GoT) results in a balanced tree and incurs $O(K + K \log K)$ steps, i.e., super-linear overhead; a branching strategy repeated $K$ times (ToT) results in $p = 2$ and incurs $O(K + K^2)$ steps, i.e., quadratic overhead.[7] Then, the total number of **reasoning steps** is $O(K + K^p)$, which is at most $O(K^2)$.*

Lemma 1 caps the worst-case *overhead* at quadratic in $K$. Together with Theorem 1, it frames the key trade-off and demonstrates the importance of the network design whose exponent $p$ stays below the decay exponent $\gamma$. In particular, XNoT adaptively decomposes tasks into the most elementary steps possible, while dynamically coordinating only the minimal dependencies required by the task. This enables XNoT to avoid the unnecessary merge complexity of tree- or graph-based strategies, achieving lower or equal reasoning overhead in common structured tasks.

**Lemma 2** (Decay vs. overhead). *Under equation 2 and overhead exponent $p$, an LLM answers a decomposed task correctly with probability $P_{\text{SPLIT}} = \exp\!\big[-aL^{\gamma}K^{\,p-\gamma}\big]$.*

Empirically, due to the super-linear decay of LLM performance with large $\gamma$ (Levy et al., 2024), decomposition often improves robustness on larger inputs, explaining the shift from monolithic toward algorithmic prompting that better supports modularization. However, existing methods differ in how they manage the recombination overhead. ToT incurs $p \approx 2$, with overhead growing rapidly as input length increases. GoT applies a merging strategy, resulting in $p \approx \log K$. In some cases, even CoT, despite not decomposing, can outperform these methods by avoiding the extra merge cost entirely. In contrast, XNoT dynamically adjusts both the number of splits $K$ and the structure of its merge operations to minimize overhead. By scaling $K$ proportionally with input length, XNoT keeps each micro-prompt within the short-input regime, reducing the impact of super-linear decay. This adaptive scaling ensures decomposition remains effective regardless of task size, enabling XNoT to maintain robust, low-error performance on both small and large instances.

---

[7]While ToT process the full query at each step, it explores partial solutions, requiring $K$ repetitions with $K$-way branching. Pruning to one branch per step still results in $O(K^2)$ evaluations, yielding quadratic overhead.

Table 2: Comparison of prompt scheme accuracy across all benchmarks with GPT-3.5-turbo.

| | keyword | | | sorting | | | set intersection | | | arithmetic | | | large-digit | | |
|---|---|---|---|---|---|---|---|---|---|---|---|---|---|---|---|
| | 5 | 10 | 20 | 16 | 32 | 64 | 32 | 64 | 128 | 8 | 16 | 32 | 8 | 16 | 32 |
| Zero Shot | 0% | 0% | 0% | 86% | 0% | 0% | 0% | 0% | 0% | 16% | 0% | 0% | 40% | 20% | 12% |
| Few Shot | 0% | 0% | 0% | 88% | 1% | 0% | 2% | 0% | 0% | 22% | 0% | 0% | 43% | 25% | 20% |
| L2M | 0% | 0% | 0% | 35% | 6% | 0% | 70% | 18% | 0% | 21% | 20% | 3% | 42% | 3% | 0% |
| SP | 2% | 0% | 0% | 74% | 0% | 0% | 19% | 2% | 1% | 34% | 30% | 2% | 22% | 19% | 6% |
| SD | 0% | 0% | 0% | 89% | 60% | 3% | 27% | 9% | 0% | 32% | 9% | 3% | 36% | 23% | 16% |
| Zero-CoT | 2% | 0% | 0% | 92% | 0% | 0% | 2% | 0% | 0% | 26% | 0% | 0% | 45% | 25% | 21% |
| PS | 0% | 0% | 0% | 0% | 0% | 0% | 4% | 0% | 0% | 20% | 1% | 0% | 18% | 16% | 8% |
| CoT | 8% | 0% | 0% | 94% | 0% | 0% | 5% | 0% | 0% | 36% | 14% | 0% | 50% | 27% | 23% |
| CoT-SC | 8% | 0% | 0% | 98% | 0% | 0% | 7% | 0% | 0% | 36% | 12% | 0% | 52% | 28% | 24% |
| AoT | 0% | 0% | 0% | 24% | 22% | 0% | 46% | 0% | 0% | 84% | 0% | 0% | 47% | 33% | 0% |
| ToT | 1% | 0% | 0% | 100% | 12% | 0% | 29% | 0% | 0% | 42% | 8% | 0% | 40% | 20% | 5% |
| ToP | 3% | 0% | 0% | 0% | 0% | 0% | 0% | 0% | 0% | 10% | 5% | 0% | 26% | 9% | 0% |
| GoT | 34% | 8% | 6% | 100% | 31% | 1% | 44% | 7% | 1% | 12% | 5% | 0% | 9% | 0% | 0% |
| **XNoT** | **93%** | **84%** | **27%** | **100%** | **92%** | **27%** | **93%** | **32%** | **20%** | **90%** | **32%** | **10%** | **98%** | **88%** | **56%** |

Table 3: Comparison of API costs, task-specific prompt character (token) count, and accuracy under GPT-4o with the smallest division of each task. While all methods achieve comparable accuracy (nea 100%), XNoT requires significantly lower API costs and manual prompt engineering effort.

| | | keyword | sorting | set intersection | arithmetic | large-digit |
|---|---|---|---|---|---|---|
| ToT | API costs | $1.041 | $0.513 | $0.320 | $0.251 | $0.292 |
| | # Characters (# Tokens) | 4986 (1274) | 3240 (2014) | 4114 (2064) | 1609 (550) | 2143 (596) |
| | Accuracy | 100% | 100% | 99% | 100% | 100% |
| GoT | API costs | $0.647 | $0.314 | $0.132 | $0.112 | $0.156 |
| | # Characters (# Tokens) | 7534 (1803) | 4640 (2534) | 5059 (2464) | 2434 (791) | 3568 (853) |
| | Accuracy | 99% | 100% | 100% | 100% | 100% |
| XNoT | API costs | **$0.526** | **$0.267** | **$0.115** | **$0.091** | **$0.085** |
| | # Characters (# Tokens) | **1033 (278)** | **1049 (352)** | **641 (209)** | **1542 (395)** | **945 (356)** |
| | Accuracy | 100% | 100% | 100% | 100% | 100% |

## 6 EXPERIMENT

**Dataset.** We evaluate the scalability of our kNoT over five benchmarks: **1) Keyword** provides an article with 5–20 sentences. **2) Sorting** provides an array of single-digit numbers with duplicates. **3) Set Intersection** provides two lists of double-digit numbers. **4) Arithmetic** provides a sequence of double-digit arithmetic. (Floating-point answers are rounded to two decimal places.) **5) Large-Digit** provides a two multi-digit numbers. We define three problem sizes for the last four use cases and prepare 100 test queries (examples shown in Appendix D.2.1) for each problem size.

**Baseline.** We compare XNoT against *Least-to-most Prompting (L2M)* (Zhou et al., 2023), *Successive Prompting (SP)* (Dua et al., 2022), *Self-discover (SD)* (Zhou et al., 2024), *Zero-CoT* (Kojima et al., 2022), *Plan-and-solve (PS)* (Wang et al., 2023), *Chain of Thought (CoT)* (Wei et al., 2022), *Self-Consistent CoT (CoT-SC)* (Wang et al., 2022), *Algorithm of Thought (AoT)* (Sel et al., 2024), *Tree of Thoughts (ToT)* (Yao et al., 2024), *Tree of Problem (ToP)* (Zebaze et al., 2024), *Graph of Thoughts (GoT)* (Besta et al., 2024a), as well as *Few-Shot* and *Zero-Shot* prompting.

**Environment.** Prompt and script design are conducted manually or with *GPT-4o* assistance. All prompt executions are conducted with *GPT-3.5-turbo*. The LLM temperature is set to 0 in all experiments to ensure deterministic and consistent outputs. Experiment results on ablation test and case studies on (5-number) Game of 24, GSM8K, and Healthcare Triage is presented in Appendix D.

### 6.1 QUANTITATIVE ANALYSIS

Table 2 summarizes the results across all benchmarks. Overall, *monolithic prompting* (Zero-CoT, CoT, CoT-SC, L2M, SP, SD, and PS) perform reasonably on small instances but consistently fail to scale as input size increases. Their linear, single-pass execution prevents modular coordination of intermediate results, limiting their scalability across tasks with larger or more complex inputs.

In contrast, *algorithmic prompting* (AoT, ToT, ToP, and GoT) introduce modular execution with tree or graph structures, improving over monolithic baselines on certain tasks. However, their reliance on rigid, hand-crafted execution scripts and fixed solution structure limits adaptability across task divisions. This reduces their practical scalability when applied to varying problem sizes.

XNoT consistently achieves stronger scalability across all tasks by generating LWT-formatted plans that adapt both the number of steps and their dependencies to the size and structure of each instance.

For **natural language tasks**, XNoT achieves the highest accuracy by decomposing articles into sentences and processing them incrementally. While GoT also performs sentence-level processing, its fixed merge strategy lacks the adaptive aggregation used by XNoT. In contrast, CoT and ToT process entire articles as single units, often failing on longer inputs due to LLM capacity limitations.

For **symbolic operation tasks**, XNoT demonstrates both step-length and step-dependency generalization by dynamically generating step-by-step plans. For example, in sorting, XNoT applies counting sort to avoid the quadratic step expansion in GoT's merge sort approach, maintaining stable performance even on larger divisions. In set intersection, XNoT checks each element individually, outperforming baselines that attempt to process all or multiple elements due to coarse splitting.

For **numerical tasks**, XNoT again outperforms baselines by generating fine-grained, sequential solution plans. While GoT performs better than monolithic methods on other tasks, its split-and-merge design is ill-suited for arithmetic and large-digit addition, where operations must be applied in strict sequence without partitioning the input. XNoT instead produces plans that apply operations step-by-step, improving reliability on longer sequences. Its flexible planning also correctly manages carry-over in large-digit addition, which other methods fail to handle effectively.

## 6.2 COST ANALYSIS

Table 3 compares XNoT with representative algorithmic prompting methods ToT and GoT in API cost, task-specific prompt size, and accuracy under GPT-4o execution. As shown, all methods achieve near 100% accuracy on the smallest division, enabling cost comparisons with no trade-offs in accuracy. XNoT achieves the lowest cost per query by leveraging short, elementary LWT-instructions that avoid redundant computation. In contrast, ToT and GoT require larger reasoning modules and repeated branching, increasing token usage and API costs. On average, XNoT reduces task-specific prompt size by 69% compared to ToT and 78% compared to GoT. This efficiency results from XNoT's intelligence amplification, generating detailed execution plans from small, reusable examples.

Note that character count alone does not fully capture API costs or total engineering effort. Both ToT and GoT perform repeated inferences within individual steps—such as branch scoring or pruning—to improve reliability, amplifying execution cost beyond what is visible from prompt size alone. In contrast, XNoT avoids such runtime repetition by performing two lightweight planning passes upfront: one to outline the solution structure, and another to compile it into a complete LWT-formatted execution plan. After this planning stage, execution proceeds in a single pass over modular, elementary steps without further branching or retries.

Moreover, ToT and GoT require significant task-specific engineering outside the prompt itself, relying on specialized Python scripts to control execution for each task *division*. These scripts are not reflected in prompt character counts but represent a substantial hidden cost in human effort and system complexity. XNoT eliminates this hidden burden by generating the full execution plan in natural language, making the process both transparent and automation-friendly.

## 7 CONCLUSION

In this work, we introduce Executable Network of Thoughts (XNoT), a prompt scheme that enables large language models to autonomously generate and execute their own solution plans. XNoT achieves strong performance across diverse reasoning tasks while reducing human effort in task-specific prompt engineering. By decomposing tasks into elementary steps, XNoT lowers LLM usage cost without sacrificing accuracy. This establishes a scalable, adaptive prompting paradigm that empowers LLMs to self-organize and execute complex reasoning processes, supporting scalable solutions to increasingly demanding real-world tasks.

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

Table 4: Terminology and Notations

| Name | Notation | Definition |
|------|----------|------------|
| Reasoning step | – | A discrete unit of inference bridging an input or intermediate result to a partial solution or next conclusion. |
| Elementary step | – | The smallest indivisible reasoning unit that transforms input into an intermediate result, e.g., summing two numbers. |
| Inference pass | – | A single forward execution of the LLM that generates output from a given prompt. |
| Modularization | – | Executing reasoning steps in separate inference passes, allowing flexible control over reasoning flow and structure. |
| Solution structure | – | A **solution structure** is the abstract outline which a prompt scheme follows to arrange reasoning steps into a chain, tree, graph, or network. *Fixed structures* are predetermined (e.g., ToT, GoT), while *dynamic structures* (e.g., XNoT) are generated at runtime per query. |
| Sequence length generalization | – | The model's ability to solve longer queries by expanding the number of reasoning steps (**step-length generalization**) while maintaining correct inter-step dependencies (**step-dependency generalization**). |
| Intelligence amplification | – | Enhancement of human reasoning by LLM assistance. XNoT achieves this by enabling the LLM to dynamically derive solution structures using a compact LWT example $\mathcal{E}$, minimizing manual prompt design. |
| Manual labor | – | The human effort needed to adapt a prompt scheme to new tasks, quantified by the character or token count of task-specific prompts. |
| LLM Workflow Template | LWT | The structured format in XNoT that labels each reasoning step and specifies its input references. Fields are dynamically replaced during execution. |
| LWT-instruction | – | A single step in a LWT script that defines an action and indexed input dependencies. |
| LWT-formatted script | $\mathcal{S}$ | A network of multiple LWT-instructions, where each instruction (node) may depend on outputs of earlier steps (edges). |
| Algorithm | $\mathcal{A}$ | The overarching procedural logic of a prompt scheme. For XNoT, see Algorithm 1. |
| Constant prompts | $\mathcal{P}_{const}$ | Prompts that encode stable task logic and remain unchanged across different queries. |
| Task-specific prompts | $\mathcal{P}_{task}$ | Prompts manually crafted for new tasks, often including examples or constraints. |
| Input query | $\mathbf{q}$ | A task instance submitted to the LLM. For generalization tasks, it often involves a variable-length sequence. |
| Task description | $\mathcal{C}$ | A brief task overview with high-level strategy hints, such as "use counting sort." |
| *Knowledge Extraction Prompt* | $\mathcal{K}$ | A knowledge extraction prompt for deriving an LLM-generated solution plan $\mathfrak{P}$ from its own latent understanding, using $\mathbf{q}$, $\mathcal{C}$, and system prompt $\mathcal{P}_{sys}$. It enforces step simplicity and acyclic positional references. |
| LLM solution plan | $\mathfrak{P}$ | A free-text reasoning outline generated by the LLM in response to $\mathcal{K}$, prior to formalization into LWT. |
| LWT example | $\mathcal{E}$ | A manually constructed LWT-formatted example on a smaller instance (e.g., a 4-number case for 8/16/32-number arithmetic), used to guide structure transfer. |
| *LWT Compilation Prompt* | $\mathcal{T}$ | A translation prompt that converts $\mathfrak{P}$ into an executable LWT $\mathcal{S}$ using $\mathcal{E}$, $\mathcal{C}$, $\mathbf{q}$, and a system prompt $\mathcal{P}_{sys}$. |

## A    EXTENDED RELATED WORK

This section provides a detailed analysis of prior prompt scheme designs, focusing on their structural assumptions, limitations, and comparisons with our proposed method, Executable Network of Thoughts (XNoT). We categorize existing approaches into two groups: *monolithic prompting*,

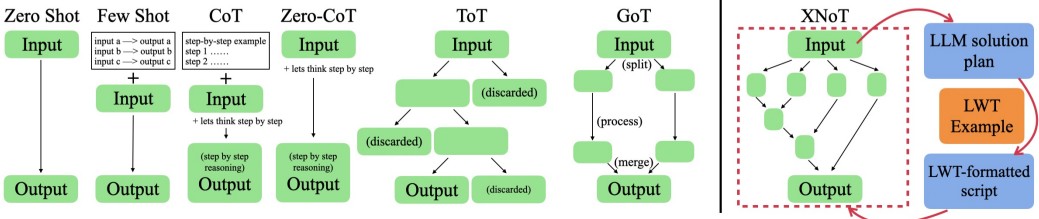

Figure 3: Additional illustration of prompt schemes.

which performs a fixed number of reasoning steps or traverses subproblems in a single loop, and *algorithmic prompting*, which defines complex solution structures such as trees or graphs to co-ordinate multi-step reasoning. We review the progression of prompt schemes, focusing on their capabilities across four key dimensions: **modularized step**, which captures the ability to isolate and execute reasoning steps separately; **elementary step**, which measures how finely the method breaks down computation; **network of thought**, which refers to the method's capacity to coordinate and reuse intermediate steps across complex dependency graphs; and **intelligence amplification**, which considers the extent to reduce human prompt engineering by leveraging the LLM's autonomous reasoning. Table 4 defines these core dimensions along with key terminology used throughout the paper.

### A.1 MONOLITHIC PROMPTING

*Chain-of-Thought (CoT)* (Wei et al., 2022) introduces a prompting strategy where LLMs are guided through reasoning using few-shot step-by-step examples and a corresponding instruction, namely, "Let's think step-by-step." However, CoT executes the entire reasoning process within a single inference pass, lacking modularization and control over the structure or flow of information. As task complexity increases, this monolithic execution leads to longer prompts and higher error rates due to the LLM's reduced reliability over extended contexts (Liu et al., 2024). While some studies (Fu et al., 2023; Jin et al., 2024) report that longer CoT outputs are associated with higher accuracy—possibly due to the implicit inclusion of more reasoning steps—they do not explicitly link this to decomposition into **elementary steps**. Crucially, CoT offers no mechanism to enforce such granularity; the LLM may ignore the example structure, usually requiring repeated sampling and answer aggregation to ensure quality (Wang et al., 2022), thereby increasing computational cost. Recent findings confirm that excessive output length ultimately degrades performance (Wu et al., 2025), highlighting the limitations of non-modular prompting.

Several extensions of CoT aim to improve its usability and consistency. *Zero-shot Chain of Thought (Zero-CoT)* (Kojima et al., 2022) shows that appending only the instruction "Let's think step by step," without the step-by-step reasoning examples, can still elicit intermediate reasoning and improved performances. *Automatic Chain-of-Thought (Auto-CoT)* (Zhang et al., 2023) leverages Zero-CoT to automatically generate step-by-step examples, increasing the portability of CoT. *Self-Consistency* (Wang et al., 2022) and follow-up works (Fu et al., 2023; Jin et al., 2024) enhance answer reliability by sampling multiple CoT results and selecting the most consistent response. Despite these improvements, all approaches inherit CoT's core limitations, including monolithic reasoning, lack of modular execution, and reliance on a single inference pass. These properties hinder scalability to complex tasks with many steps, where context length becomes a bottleneck (Wu et al., 2025). As such, CoT-style methods are also criticized for their weak planning capabilities (Stechly et al., 2024).

Subsequent work moves toward modularized prompting, either by introducing structured intermediate stages that impose planning behaviors (Wang et al., 2023; Zhou et al., 2024), or by iterating over a decomposed sequence of subproblems (Zhou et al., 2023; Dua et al., 2022; Ning et al., 2023). To improve planning ability, *Plan-and-Solve Prompting (PSP)* (Wang et al., 2023) adopts a monolithic format similar to Zero-CoT but introduces more explicit prompts for planning. *Self-Discover (SD)* (Zhou et al., 2024) advances this idea with prompt modules that offer conceptual heuristics. SD applies three separate inference steps: selecting a planning module, adapting it to the query, and

executing the adapted plan. Abstraction-of-Thought (Hong et al., 2024) simply provides slightly more advanced examples to CoT, where the step-by-step process involves abstractions and high-level ideas. Nevertheless, these methods remain rigid, relying on handcrafted scripts and lacking runtime structural adaptability.

*Least-to-Most Prompting (L2M)* (Zhou et al., 2023) uses a single LLM to solve subproblems with increasing difficulty, while *Successive Prompting (SP)* (Dua et al., 2022) alternates between two (separately fine-tuned) LLM models to iteratively decompose and solve subproblems. Both approaches require extensive manual configuration tailored to each decomposition stage (e.g., examples for first-stage decomposition versus later stages conditioned on earlier subproblem solutions). Moreover, as they accumulate intermediate outputs in the prompt, context length grows linearly with the number of steps, eventually encountering the same limitations as CoT. *Skeleton-of-Thought (SoT)* (Ning et al., 2023) proposes a two-stage scheme to first generate an outline, then execute each point with separate inference calls. This strategy achieves significant speedups with parallel inferences. However, SoT lacks inter-step coherence and ignores the dependencies between points, thus limiting its applicability to tasks requiring sequential or interdependent reasoning, such as math or logic.

Overall, monolithic prompting methods lack modularity, operate at coarse granularity, and fail to support alternative solution structures automatically. Their performance degrades with increasing input complexity or sequence length. On the contrary, XNoT retains the simple, few-step prompting scheme for planning and a sequential execution, consistent with the simplicity of monolithic prompting, yet forms a structured network of dependencies between **modularized** reasoning steps via indexed fields through the *LWT format*. The indexing design further enables precise reference to subcomponents of the input, allowing each instruction to be tailored to an **elementary** reasoning step. As a result, XNoT achieves fine-grained decomposition, modular execution, and network of thought generalization within a linear control flow.

### A.2 ALGORITHMIC PROMPTING

Prompt schemes in this category leverage complex algorithm designs to offer modular LLM execution and introduce alternative solution structures that arrange reasoning steps following tree or graph-based abstractions. For instance, *Tree-of-Thoughts (ToT)* (Yao et al., 2024) employs a breadth-first search (BFS) algorithm, where the tree search branches out by repeatedly prompting LLMs to generate the next step, then pruned by prompting the LLMs to evaluate the resulting state. *Algorithm-of-Thought (AoT)* (Sel et al., 2024) takes a step beyond ToT to include depth-first search (DFS) with extensive prompt and code engineering. *Forest-of-Thought (FoT)* (Bi et al., 2024) duplicates ToT by maintaining parallel ToT trees as separate branches, and iteratively selecting the tree to continue branching to the next level by scoring with LLMs the intermediate states and comparing across different trees.

*Graph-of-Thoughts (GoT)* (Besta et al., 2024a) generalize ToT's approach with an additional "merge" module, which allows different branches to be joined back together. However, it only provides use cases that are all solved by a single split-solve-then-merge workflow. In particular, to enhance performance, GoT needs to repeat a particular action multiple times and then select the best result by external or LLM evaluation. *Tree-of-Problems (ToP)* (Zebaze et al., 2024) is similar to GoT but removes the repeated inference and selection process to reduce complexity. In addition, while GoT focuses on dissecting an input query into subproblems, Branch-solve-merge (Saha et al., 2024) follows the same structure but targets open-ended problems, aiming to evaluate different LLM responses over several predefined evaluation criteria branches, and merging all evaluations for a final score.

Although algorithmic prompting often achieves stronger performance, it typically requires task-specific logic and prompt engineering, limiting adaptability. First, these methods are difficult to generalize across tasks. For example, both ToT (Yao et al., 2024) and GoT (Besta et al., 2024a) rely on separate Python files with hard-coded logic tailored to each task, requiring manual implementation of task-specific reasoning flows. By contrast, XNoT operates with a simple loop structure similar to monolithic prompting methods, but incorporates more expressive networked dependencies through the LWT format.

Second, despite their modular design, algorithmic prompting methods are brittle due to reliance on handcrafted code and static prompt templates. They cannot dynamically and automatically adjust

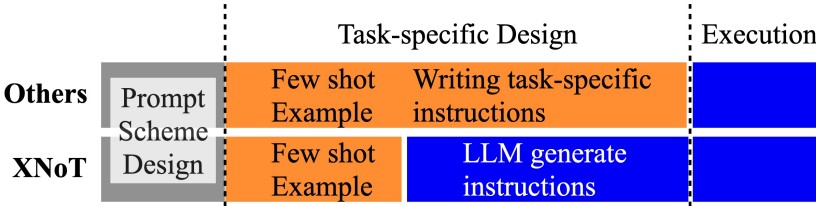

Figure 4: Illustrations of human labor (orange block ■) and LLM operation (blue block ■) required for adapting a prompt scheme to a new task. prompt engineering algorithm. Gray blocks ■ indicate the labor involved in designing constant prompts, which do not require redesign for new tasks.

Table 5: Delineation of the system versus the task-specific prompts (an average character count) for XNoT. We highlight constant, task-specific, and LLM-generated prompts in gray, orange, and blue.

| Stage | Constant Prompt $\mathcal{P}_{const}$ | Task-specific Prompt $\mathcal{P}_{task}$ |
|-------|---------------------------------------|-------------------------------------------|
| I     | Extraction instructions (274)         | Task description $C$ (172)                 |
| II    | Translation instructions (516)        | LWT Example $\mathcal{E}$ (561)            |
| III   | –                                     | LWT-formatted script $\mathcal{S}$ (1767)  |

their solution structures at runtime and lack the flexibility to generalize across varying input lengths within the same task, failing to preserve **elementary steps**. For example, ToT is fixed to the 4-number version of the Game of 24 and cannot be applied to 5-number inputs without substantial redesign. GoT, in turn, requires different prompt examples for different sequence lengths, such as in sorting tasks, further limiting scalability.

**Remark.** A primary limitation of ToT and GoT is their insufficient support for *elementary steps* and rigid structure for *networks of thought*. ToT processes the entire query, branching on partial solutions and discarding incorrect ones, while GoT imposes a predefined structure, e.g., a four-way split-then-merge graph. Although four-way splitting can reduce an 8-number addition to 2-number operations, it forces each step of a 64-number addition to handle eight numbers in zero-shot, compromising elementary steps and degrading performance (Table 2). Simply increasing a four-way split to a 32-way split doesn't address how to merge and chain the resulting partial outputs; while merging four results may be straightforward, merging 32 is significantly more complex. In contrast, XNoT applies *intelligence amplification* (not supported by prior works (Yao et al., 2024; Besta et al., 2024a)) to generate a dynamic structure for each query. By splitting the query at the outset, XNoT employs *elementary steps* for precise execution. XNoT achieves both *step-length generalization* by dynamically adding more steps for larger problems and *step-dependency generalization* through its flexible indexed input fields that maintain correct relationships between operations regardless of problem size. The LWT format's ability to specify precise dependencies through indexing enables XNoT to organize complex reasoning networks that adapt to varying problem sizes more effectively than prior approaches that process the entire query (CoT, ToT) or coarse partitions (GoT), thus improving reliability across varying problem sizes.

Moreover, XNoT leverages **intelligence amplification** to generalize a single design across a range of input complexities. A single LWT example for an arithmetic sequence of length four can be reused to solve instances of length 8, 16, or 32. XNoT dynamically constructs a query-specific LWT-formatted script by following the structural logic and syntax of the example, scaling the solution to match the input while maintaining modularity and step granularity.

## A.3 OVERALL COMPARISON

In contrast to prior work, XNoT achieves full coverage across all four dimensions: modularization, elementary step decomposition, networked reasoning, and intelligence amplification (see Table 1). It requires no handcrafted control logic and supports structural generalization through reusable prompting. As shown in Section 6, XNoT consistently outperforms baseline methods, especially in tasks requiring long-sequence processing or dynamic decomposition. For example, a script for a

4-number version of Game of 24 can be generalized by the LLM into a 5-number version using a single prompt. This level of structural transfer and modular reuse is not supported by any existing method.

Furthermore, we specifically highlight a comparison between XNoT and prior works on the human effort in prompt engineering. First, we present an illustration that demonstrates how XNoT solves the Prompt Engineering Problem with less human effort. As shown in Figure 4, unlike prior approaches that require redesigning both task instructions and few-shot examples for each task, XNoT only reduces the required manual effort for prompt design by allowing LLM operations to provide the task instructions within the task-specific design. The second phase usually must be repeated for each task $t \in \mathcal{T}$, creating a bottleneck for scalability. For example, ToT and GoT require separate manual instructions and few-shot examples for multiple modules. In contrast, XNoT reduces human effort by leveraging LLMs to automatically generate suitable instructions.

Besides, Table 5 shows a breakdown of the constant and task-specific prompt components in XNoT. Constant prompts, which remain unchanged across tasks, include extraction and translation instructions. Task-specific prompts consist of a brief task description, a representative example, and an LLM-generated script. The average character counts indicate that most of the prompt volume resides in the automatically generated content, further reducing the human burden in prompt engineering.

## A.4   OTHER WORKS

Beyond intrinsic prompting strategies, a variety of alternative approaches have been explored to enhance LLM reasoning. Nevertheless, they fall outside the scope of this work since they require fine-tuning the underlying LLMs or external tools or interactive environments (Yang et al., 2018; Shridhar et al., 2021).

Reactive prompting methods, such as ReAct (Yao et al., 2022) and Reflexion (Shinn et al., 2024), and ExpeL (Zhao et al., 2024), interleave LLM reasoning with feedback from an external environment (Yang et al., 2018; Shridhar et al., 2021) to gain experience. These techniques are particularly effective for embodied agents and sequential decision-making tasks where intermediate actions can be evaluated and corrected. However, they do not provide a structured decomposition of reasoning steps and are not applicable in static settings like symbolic computation or pure language understanding, where no interactive feedback loop exists.

Tool-use prompting introduces external computation resources into the reasoning pipeline. Program-aided methods (Chen et al., 2023b; Gao et al., 2023), for instance, offload symbolic subroutines to interpreters or APIs. While highly precise, these approaches are generally limited to tasks with formal programmatic definitions and are unsuitable for broader language-based reasoning or general-purpose workflows. Moreover, tool integration introduces operational overhead and additional infrastructure requirements.

Open-ended prompting methods aim to enhance LLMs' reasoning through abstraction, internal dialogue, or collaboration. For example, Diagram-of-Thought (DoT) (Zhang et al., 2024b) models the reasoning process as an internal directed acyclic graph within a single LLM, organizing sub-problems through refinement and verification cycles. These frameworks aim to improve expressivity and creativity but often lack structured modularization and introduce complexity in control flow and evaluation.

Finally, some approaches involve modifying the LLM itself through fine-tuning or reinforcement learning (Lester et al., 2021; Li and Liang, 2021; Hu et al., 2022). While these methods can enhance reasoning performance, they incur high computational costs and reduce portability across tasks and model architectures. In parallel, retrieval-augmented generation (RAG) (Lewis et al., 2020; Asai et al., 2024; Zhang et al., 2024a) improves factual accuracy by supplying the LLM with external information retrieved based on the input query. However, RAG requires a comprehensive data source, which may limit its applicability across diverse or under-resourced tasks.

## A.5   RELATION TO MULTI-AGENT SYSTEMS (MAS)

XNoT is a *prompt-native, declarative* framework that encodes reasoning as an explicit DAG within the prompt via *LLM Workflow Template (LWT)*. LWT captures decomposition, control flow, and

Table 6: Conceptual comparison across paradigms.

| Aspect | MAS (Multi-Agent Systems) | XNoT | Algorithmic Prompting | Monolithic Prompting |
|---|---|---|---|---|
| Primary strength | Coordination of roles and external tools (memory, retrieval, APIs). | Prompt-level DAG (LWT) enabling long-horizon reasoning. | Fixed structures (trees, split–merge) for modularization. | Simple linear chains; efficient but brittle on long dependencies. |
| Typical domains | Software engineering, social simulation, interactive planning. | Decomposable tasks with fixed solution shapes. | Decomposable tasks with fixed solution shapes. | Short or open-ended prompts without control flow. |
| Cost profile | High (orchestration, memory, retrieval, tools, planners). | Low (single LLM; no external infrastructure). | Low–medium (multiple passes, prompt design). | Lowest (single pass). |
| Reasoning pattern | Parallel/linear orchestration across roles. | Versatile Network-of-Thought | Hard-coded tree or split–merge; limited adaptivity. | Linear chain-of-thought; no branching. |
| Main challenge | Coordination overhead, conflict resolution. | Maintaining coherence over long, interdependent DAGs. | Inflexibility and per-task engineering. | Error accumulation, limited generalization. |

message dependencies, enabling end-to-end execution by a single model (optionally distributed) without role dispatch. See Table 6 for a side-by-side comparison of XNoT with MAS and related prompting paradigms.

By contrast, Multi-agent Systems (MAS) (Wu et al., 2023; Qian et al., 2024; Liu et al., 2025) organizes fixed or dynamic *roles* (planner, executor, validator) and coordinates them via message passing and tools (Wu et al., 2023; Qian et al., 2024), emphasizing system-level orchestration for domains like software development (Qian et al., 2024; Islam et al., 2024), social simulation (Park et al., 2023; Yin et al., 2023), and interactive planning (Hao et al., 2025; Zhang et al., 2025). These settings are largely orthogonal to prompt-native, language-only reasoning.

Because LWT materializes a DAG, XNoT can serve as a *control backbone* within MAS controllers (principled sub-task routing, auditable traces, optional parallelization). Thus the two are complementary: MAS handles *system-level* coordination, while XNoT provides an interpretable, low-overhead reasoning core for language-only workflows.

## B  XNoT Design and Implementation Details

We present detailed examples of XNoT, providing constant prompt templates, i.e., *Knowledge Extraction Prompt* and *LWT Compilation Prompt*, and LLM-generated prompts, i.e., the LLM solution plan and the LWT-formatted script. Please see Appendix D.2.1 for input query examples. Note that occurrences of "..." in the prompts are part of the actual prompt text rather than abbreviations, whereas "=====" denotes omitted LLM-generated outputs that obviously follow a repetitive pattern.

### B.1  Full prompt examples with 8-number arithmetic calculation

**Knowledge Extraction Prompt**  Text in red and orange corresponds to the input query **q** and the task description $C$, respectively, and can be swapped out for other use-case examples. In the following prompt examples, we follow the color scheme where text in black font color denotes constant prompts that remain unchanged across different tasks, text in orange denotes human-prepared task-specific prompts, and text in blue denotes LLM-generated prompts.

---

*Given the following question:*
*Input: 5\*5/5\*4+8-8+3\*9*

---

*Context: Perform the arithmetic result of Input. You can only operate two numbers at a time. Calculate from left to right. Do multiplication and division first.*
*The Input is the input query. The Context is the goal we want to achieve.*
*Please use your knowledge to create a solution in a step-by-step manner without any numbers.*
*Every step needs to be as easy as possible. Use Step0, Step1, Step2 to represent the result.*
*Don't use a for loop to reduce the step. Don't directly use any element in the Input.*

**LWT Compilation Prompt**   Text in blue at the top is the LLM solution plan, whereas text in orange at the bottom is the LWT example. Note that parts of the prompt (Input and Context) are identical to the *Knowledge Extraction Prompt*.

*Based on your expert knowledge:*
*Step0: first number * second number*
*Step1: Step0 / third number*
*Step2: Step1 * fourth number*
*Step3: Step2 + fifth number*
*Step4: Step3 - sixth number*
*Step5: Step4 + seventh number * eighth number*
*and the example below, create a script to solve the following question:*
*Input: 5*5/5*4+8-8+3*9*
*Context: Perform the arithmetic result of Input. You can only operate two numbers at a time. Calculate from left to right. Do multiplication and division first.*
*The Input is the input query. The Context is the goal we want to achieve.*
*You have to follow the rules to create a script.*
*This script should be numbered and contains several instructions to be called line-by-line in a sequential order.*
*Use (number) to represent each line. The line numbering starts from 0.*
*You can use LLM Inference: use LLM("Your Instruction") to find the answer.*
*Use {(index)} to represent the variable you want to replace with the previous result.*
*Use {(Input)}, {(Set1)}, ... to represent Input, not allowed to use numbers directly.*
*Use Python indexing to get the element in the list (E.g., {(0)}[0], {(0)}[1]). Do not directly use numbers.*
*Here is one example.*
*(0)=LLM("Given {(input)}, Split the numbers without operators. Only output list.")*
*(1)=LLM("Add({(0)}[0], {(0)}[2]). Only output number. If contains floating point, round to two decimal places.")*
*(2)=LLM("Subtraction({(0)}[1], {(1)}). Only output number. If contains floating point, round to two decimal places.")*
*(3)=LLM("Multiply({(0)}[1], {(1)}). Only output number. If contains floating point, round to two decimal places.")*
*(4)=LLM("Divide({(0)}[1], {(1)}). Only output number. If it contains a floating point, round to two decimal places.")*

**LWT-formatted script:**

*(0)=LLM("Given {(input)}, Split the numbers without operators. Only output list.")*
*(1)=LLM("Multiply({(0)}[0], {(0)}[1]). Only output number. If contains floating point, round to two decimal places.")*
*(2)=LLM("Divide({(1)}, {(0)}[2]). Only output number. If contains floating point, round to two decimal places.")*
*(3)=LLM("Multiply({(2)}, {(0)}[3]). Only output number. If contains floating point, round to two decimal places.")*
*(4)=LLM("Add({(3)}, {(0)}[4]). Only output number. If contains floating point, round to two decimal places.")*
*(5)=LLM("Subtraction({(4)}, {(0)}[5]). Only output number. If contains floating point, round to two decimal places.")*

> *(6)=LLM("Multiply({(0)}[6], {(0)}[7]). Only output number. If contains floating point, round to two decimal places.")*
> *(7)=LLM("Add({(5)}, {(6)}). Only output number. If it contains a floating point, round to two decimal places.")*

## B.2  TASK-SPECIFIC PROMPTS AND LLM-GENERATED PROMPTS OF EACH USE CASE

**Keyword Counting**   Every Input in keyword counting contains an article with 14 to 20 sentences. XNoT divide the whole article into sentences and check the keyword one by one. In the final step, we will output an array combined with previous results.

**Task description:**

> *Output all words about countries in the article. You can separate the article into sentences first.*

**LWT example:**

> *(0)=LLM("Split the following article into sentences: '{(input)}'. Output an array.")*
> *(1)=LLM("Extract all country names (no continents) in the order of their appearance from the following sentence (repeated is allowed): "{(0)}[0]" Output [] if not exist any country.")*
> *(2)=LLM("Extract all country names (no continents) in the order of their appearance from the following sentence (repeated is allowed): "{(0)}[1]" Output [] if not exist any country.")*
> *(3)=LLM("Extract all country names (no continents) in the order of their appearance from the following sentence (repeated is allowed): "{(0)}[2]" Output [] if not exist any country.")*
> *...*
> *(20)=LLM("Extract all country names (no continents) in the order of their appearance from the following sentence (repeated is allowed): "{(0)}[19]" Output [] if not exist any country.")*
> *(21)=LLM("Combine {(1)}, {(2)}, {(3)}, {(4)}, {(5)}, {(6)}, {(7)}, {(8)}, {(9)}, {(10)}, {(11)}, {(12)}, {(13)}, {(14)}, {(15)}, {(16)}, {(17)}, {(18)}, {(19)}, {(20)} in one array. Repeated is allowed.")*

**LLM solution:**

> *Step0: Identify the first sentence in the Input.*
> *Step1: Extract the country names from the first sentence: Norway, Sweden, Norway, Sweden.*
> *Step2: Identify the second sentence in the Input.*
> *Step3: Extract the country names from the second sentence: Canada, United States.*
> *Step4: Identify the third sentence in the Input.*
> *Step5: Extract the country names from the third sentence: Canada, United States.*
> *Step6: Identify the fourth sentence in the Input.*
> *Step7: Extract the country names from the fourth sentence: Portugal, Spain, Portugal.*
> *Step8: Identify the fifth sentence in the Input.*
> *Step9: Extract the country names from the fifth sentence: Spain, Portugal, Norway, Sweden, Canada, United States.*
> *Step10: Identify the sixth sentence in the Input.*
> *Step11: Extract the country names from the sixth sentence: South Korea, North Korea.*
> *Step12: Identify the seventh sentence in the Input.*
> *Step13: Extract the country names from the seventh sentence: North Korea, South Korea.*
> *Step14: Identify the eighth sentence in the Input.*
> *Step15: Extract the country names from the eighth sentence: Australia, New Zealand.*
> *Step16: Identify the ninth sentence in the Input.*
> *Step17: Extract the country names from the ninth sentence: Australia, New Zealand.*
> *Step18: Identify the tenth sentence in the Input.*
> *Step19: Extract the country names from the tenth sentence: South Africa, Zimbabwe.*
> *Step20: Identify the eleventh sentence in the Input.*
> *Step21: Extract the country names from the eleventh sentence: South Africa, Zimbabwe.*

*Step22: Identify the twelfth sentence in the Input.*
*Step23: Extract the country names from the twelfth sentence: Norway, Australia, Canada, Zimbabwe.*

**LWT-formatted script:**

*(0)=LLM("Split the following article into sentences: '{(input)}'. Output an array.")*
*(1)=LLM("Extract all country names (no continents) in the order of their appearance from the following sentence (repeated is allowed): '{(0)}[0]'. Output [] if not exist any country.")*
*(2)=LLM("Extract all country names (no continents) in the order of their appearance from the following sentence (repeated is allowed): '{(0)}[1]'. Output [] if not exist any country.")*
*(3)=LLM("Extract all country names (no continents) in the order of their appearance from the following sentence (repeated is allowed): '{(0)}[2]'. Output [] if not exist any country.")*
*(4)=LLM("Extract all country names (no continents) in the order of their appearance from the following sentence (repeated is allowed): '{(0)}[3]'. Output [] if not exist any country.")*
*(5)=LLM("Extract all country names (no continents) in the order of their appearance from the following sentence (repeated is allowed): '{(0)}[4]'. Output [] if not exist any country.")*
*(6)=LLM("Extract all country names (no continents) in the order of their appearance from the following sentence (repeated is allowed): '{(0)}[5]'. Output [] if not exist any country.")*
*(7)=LLM("Extract all country names (no continents) in the order of their appearance from the following sentence (repeated is allowed): '{(0)}[6]'. Output [] if not exist any country.")*
*(8)=LLM("Extract all country names (no continents) in the order of their appearance from the following sentence (repeated is allowed): '{(0)}[7]'. Output [] if not exist any country.")*
*(9)=LLM("Extract all country names (no continents) in the order of their appearance from the following sentence (repeated is allowed): '{(0)}[8]'. Output [] if not exist any country.")*
*(10)=LLM("Extract all country names (no continents) in the order of their appearance from the following sentence (repeated is allowed): '{(0)}[9]'. Output [] if not exist any country.")*
*(11)=LLM("Extract all country names (no continents) in the order of their appearance from the following sentence (repeated is allowed): '{(0)}[10]'. Output [] if not exist any country.")*
*(12)=LLM("Combine {(1)}, {(2)}, {(3)}, {(4)}, {(5)}, {(6)}, {(7)}, {(8)}, {(9)}, {(10)}, {(11)} in one array. Repeated is allowed.")*

## Sorting    Task description:

*Sort input in ascending order. You can use counting sort.*

## LWT example:

*(0)=LLM("Initialize an array of size 10 to zero.")*
*(1)=LLM("Increment the count at index {(input)}[0] in {(0)} (index start from 0). Only output updated array.")*
*(2)=LLM("Increment the count at index {(input)}[1] in {(1)} (index start from 0). Only output updated array.")*
*...*
*(16)=LLM("Increment the count at index {(input)}[15] (start from 0) in {(15)}. Only output updated array.")*
*...*
*(length+1)=LLM("Convert {(length)} in English. Output an array.")*
*(length+2)=LLM("The array should contain {(length+1)}[0] 0s, {(length+1)}[1] 1s, {(length+1)}[2] 2s, {(length+1)}[3] 3s, {(length+1)}[4] 4s. Output in array format.")*
*(length+3)=LLM("The array should contain {(length+1)}[5] 5s, {(length+1)}[6] 6s, {(length+1)}[7] 7s, {(length+1)}[8] 8s, {(length+1)}[9] 9s. Output in array format.")*
*(length+4)=LLM("Combine {(length+2)} and {(length+3)} in ascending order. Only output array.")*

## LLM plan:

*Step0: Start with the input list: ['first number', 'second number', 'third number', 'fourth number', 'fifth number', 'sixth number', 'seventh number', 'eighth number', 'ninth number', 'tenth number', 'eleventh number', 'twelfth number', 'thirteenth number', 'fourteenth number', 'fifteenth number', 'sixteenth number'].*
*Step1: Identify the smallest number in the list and place it at the beginning of a new list.*
*Step2: Identify the next smallest number in the original list and place it after the first number in the new list.*
*Step3: Continue this process, each time finding the next smallest number from the original list and adding it to the new list.*
*Step4: Repeat this process until all numbers from the original list are placed in the new list in ascending order.*
*Step5: The new list now contains the numbers sorted in ascending order: ['fifth number', 'seventh number', 'fourteenth number', 'first number', 'sixth number', 'second number', 'third number', 'eighth number', 'fifteenth number', 'sixteenth number', 'ninth number', 'eleventh number', 'twelfth number', 'thirteenth number', 'tenth number', 'fourth number'].*

**LWT-formatted script:**

*(0)=LLM("Initialize an array of size 10 to zero.")*
*(1)=LLM("Increment the count at index {(input)}[0] in {(0)} (index start from 0). Only output updated array.")*
*(2)=LLM("Increment the count at index {(input)}[1] in {(1)} (index start from 0). Only output updated array.")*
*(3)=LLM("Increment the count at index {(input)}[2] in {(2)} (index start from 0). Only output updated array.")*
*(4)=LLM("Increment the count at index {(input)}[3] in {(3)} (index start from 0). Only output updated array.")*
*(5)=LLM("Increment the count at index {(input)}[4] in {(4)} (index start from 0). Only output updated array.")*
*================================================*
*(15)=LLM("Increment the count at index {(input)}[14] in {(14)} (index start from 0). Only output updated array.")*
*(16)=LLM("Increment the count at index {(input)}[15] in {(15)} (index start from 0). Only output updated array.")*
*(17)=LLM("Convert {(16)} in English. Output an array.")*
*(18)=LLM("The array should contain {(17)}[0] 0s, {(17)}[1] 1s, {(17)}[2] 2s, {(17)}[3] 3s, {(17)}[4] 4s. Output in array format.")*
*(19)=LLM("The array should contain {(17)}[5] 5s, {(17)}[6] 6s, {(17)}[7] 7s, {(17)}[8] 8s, {(17)}[9] 9s. Output in array format.")*
*(20)=LLM("Combine {(18)} and {(19)} in ascending order. Only output array.")*

**Set Operation    Task description:**

*Find the intersection of two input. You can check every element in set1 one by one.*

**LWT example:**

*(0)=LLM("Find the intersection for [{(Set1)}[0]] and {(Set2)}. Output [] if mutually exclusive.")*
*(1)=LLM("Find the intersection for [{(Set1)}[1]] and {(Set2)}. Output [] if mutually exclusive.")*
*...*
*(length-1)=LLM("Find the intersection for [{(Set1)}[length-1]] and {(Set2)}. Output [] if mutually exclusive.")*
*(length)=LLM("Combine (0), (1), (2), ... ,{(length-1)} in one array.")*

**LLM plan:**

> *Step0: Start with the first number in Set1. Check if this number is in Set2. If it is, include it in the result.*
> *Step1: Move to the second number in Set1. Check if this number is in Set2. If it is, include it in the result.*
> *Step2: Move to the third number in Set1. Check if this number is in Set2. If it is, include it in the result.*
> *Step3: Move to the fourth number in Set1. Check if this number is in Set2. If it is, include it in the result.*
> *Step4: Move to the fifth number in Set1. Check if this number is in Set2. If it is, include it in the result.*
> *Step5: Move to the sixth number in Set1. Check if this number is in Set2. If it is, include it in the result.*
> *Step6: Move to the seventh number in Set1. Check if this number is in Set2. If it is, include it in the result.*
> *Step7: Move to the eighth number in Set1. Check if this number is in Set2. If it is, include it in the result.*
> *===================================================*
> *Step29: Move to the thirtieth number in Set1. Check if this number is in Set2. If it is, include it in the result.*
> *Step30: Move to the thirty-first number in Set1. Check if this number is in Set2. If it is, include it in the result.*
> *Step31: Move to the thirty-second number in Set1. Check if this number is in Set2. If it is, include it in the result.*

**LWT-formatted script:**

> *(0)=LLM("Find the intersection for [{(Set1)}[0]] and {(Set2)}. Output [] if mutually exclusive.")*
> *(1)=LLM("Find the intersection for [{(Set1)}[1]] and {(Set2)}. Output [] if mutually exclusive.")*
> *(2)=LLM("Find the intersection for [{(Set1)}[2]] and {(Set2)}. Output [] if mutually exclusive.")*
> *(3)=LLM("Find the intersection for [{(Set1)}[3]] and {(Set2)}. Output [] if mutually exclusive.")*
> *(4)=LLM("Find the intersection for [{(Set1)}[4]] and {(Set2)}. Output [] if mutually exclusive.")*
> *(5)=LLM("Find the intersection for [{(Set1)}[5]] and {(Set2)}. Output [] if mutually exclusive.")*
> *(6)=LLM("Find the intersection for [{(Set1)}[6]] and {(Set2)}. Output [] if mutually exclusive.")*
> *===================================================*
> *(29)=LLM("Find the intersection for [{(Set1)}[29]] and {(Set2)}. Output [] if mutually exclusive.")*
> *(30)=LLM("Find the intersection for [{(Set1)}[30]] and {(Set2)}. Output [] if mutually exclusive.")*
> *(31)=LLM("Find the intersection for [{(Set1)}[31]] and {(Set2)}. Output [] if mutually exclusive.")*
> *(32)=LLM("Combine {(0)}, {(1)}, {(2)}, {(3)}, {(4)}, {(5)}, {(6)}, {(7)}, {(8)}, {(9)}, {(10)}, {(11)}, {(12)}, {(13)}, {(14)}, {(15)}, {(16)}, {(17)}, {(18)}, {(19)}, {(20)}, {(21)}, {(22)}, {(23)}, {(24)}, {(25)}, {(26)}, {(27)}, {(28)}, {(29)}, {(30)}, {(31)} in one array.")*

**Large Number Addition    Task description:**

*Calculate the result of the input. You can add one digit to a one-digit starting from the least significant digit.*

**LWT example:**

*(0)=LLM("Split {(input)} by + and output in string format in an array.")*
*(1)=LLM("Calculate {(0)}[0][15]+{(0)}[1][15]. Only output result.")*
*(2)=LLM("Calculate {(1)} divide 10, Only output integer.")*
*(3)=LLM("Calculate {(2)}+{(0)}[0][14]+{(0)}[1][14]. Only output result.")*
*(4)=LLM("Calculate {(3)} divide 10, Only output integer.")*
*(5)=LLM("Calculate {(4)}+{(0)}[0][13]+{(0)}[1][13]. Only output result.")*
*(6)=LLM("Calculate {(5)} divide 10, Only output integer.")*

*......*
*(2\*length-1)=LLM("Calculate {(2\*length-2)}+{(0)}[0][0]+{(0)}[1][0]. Only output result.")*
*(2\*length)=LLM("Calculate {(2\*length-1)} divide 10, Only output integer.")*
*(2\*length+1)=LLM("Convert to integer: {(2\*length)}{(2\*length-1)}[-1]{(2\*length-3)}[-1]{(2\*length-5)}[-1]......{(7)}[-1]{(5)}[-1]{(3)}[-1]{(1)}[-1]")*

**LLM plan:**

*Step0: Start with the least significant digits of the first number and the second number. Add them together.*
*Step1: Move to the next least significant digits of the first number and the second number. Add them together, including any carry from Step0.*
*Step2: Move to the next least significant digits of the first number and the second number. Add them together, including any carry from Step1.*
*Step3: Move to the next least significant digits of the first number and the second number. Add them together, including any carry from Step2.*
*Step4: Move to the next least significant digits of the first number and the second number. Add them together, including any carry from Step3.*
*Step5: Move to the next least significant digits of the first number and the second number. Add them together, including any carry from Step4.*
*Step6: Move to the next least significant digits of the first number and the second number. Add them together, including any carry from Step5.*
*Step7: Move to the most significant digits of the first number and the second number. Add them together, including any carry from Step6.*
*Final Result: Combine all the results from Step0 to Step7 to get the final sum.*

**LWT-formatted script:**

*(0)=LLM("Split '{(input)}' by + and output in string format in an array.")*
*(1)=LLM("Calculate {(0)}[0][-1]+{(0)}[1][-1]. Only output result.")*
*(2)=LLM("Calculate {(1)} divide 10, Only output integer.")*
*(3)=LLM("Calculate {(2)}+{(0)}[0][-2]+{(0)}[1][-2]. Only output result.")*
*(4)=LLM("Calculate {(3)} divide 10, Only output integer.")*
*(5)=LLM("Calculate {(4)}+{(0)}[0][-3]+{(0)}[1][-3]. Only output result.")*
*(6)=LLM("Calculate {(5)} divide 10, Only output integer.")*
*(7)=LLM("Calculate {(6)}+{(0)}[0][-4]+{(0)}[1][-4]. Only output result.")*
*(8)=LLM("Calculate {(7)} divide 10, Only output integer.")*
*(9)=LLM("Calculate {(8)}+{(0)}[0][-5]+{(0)}[1][-5]. Only output result.")*
*(10)=LLM("Calculate {(9)} divide 10, Only output integer.")*
*(11)=LLM("Calculate {(10)}+{(0)}[0][-6]+{(0)}[1][-6]. Only output result.")*
*(12)=LLM("Calculate {(11)} divide 10, Only output integer.")*
*(13)=LLM("Calculate {(12)}+{(0)}[0][-7]+{(0)}[1][-7]. Only output result.")*
*(14)=LLM("Calculate {(13)} divide 10, Only output integer.")*

> *(15)=LLM("Calculate {(14)}+{(0)}[0][-8]+{(0)}[1][-8]. Only output result.")*
> *(16)=LLM("Calculate {(15)} divide 10, Only output integer.")*
> *(17)=LLM("Convert to integer: {(16)}{(15)}[-1]{(13)}[-1]{(11)}[-1]{(9)}[-1]{(7)}[-1]{(5)}[-1]{(3)}[-1]{(1)}[-1]")*

### B.3 PROMPTS AND LLM-GENERATED PROMPTS FOR THE GAME OF 24

For the Game of 24, the planner LLM does not inherently possess the symbolic reasoning ability required to decompose the task procedurally. To address this, we modify the roles of Stage 1 and Stage 2. In the adapted setup, Stage 1 immediately works with the LWT example and generates a candidate LWT-formatted script without access to a specific input query. Given the complexity of the task, it is challenging to constrain the LLM to output only the structured script format in this step, as the model tends to include explanations or extra text. Thus, Stage 2 takes the Output of Stage 1 and extracts the final, clean LWT-formatted script. Notably, we perform Stage 1 and Stage 2 only once, rather than per input query, as the resulting plan is reusable across all instances. This adaptation allows XNoT to support 5-number variants of the Game of 24, which prior methods cannot accommodate due to their rigid control logic and limited flexibility.

**Knowledge Extraction Prompt (Stage 1)**

> *Given the following question:* [task description]
> *And a solution structure example for four numbers:* [LWT example]
> *Please use your knowledge to create a solution structure for five numbers*
> *CAREFULLY CHECK EVERYSTEP. MAKE SURE IT HAS THE INPUT FIELD* {(N)} *IT NEEDS TO REFERENCE OUTPUT FROM A PREVIOUS STEP. Make sure it is in the correct syntax. E.g., (from (0)) should be* {(0)}*! MAKE SURE EACH STEP REFERENCES THE CORRESPONDING STEP, FOLLOWING THE SOLUTION STRUCTURE FOR FOUR NUMBERS. MAKE SURE THE EXAMPLE IN THE INSTRUCTION ARE UPDATED FROM FOUR NUMBER TO FIVE NUMBER CASE, I.E. ADD ONE MORE NUMBER TO THE EXAMPLE, e.g., [2 × 3 4 8 — 6 4 8] becomes [2 × 3 4 8 1 — 6 4 8 1]*

**LWT Compilation Prompt (Stage 2)**

> *You have to create a script to solve Game of 24 for five numbers.*
> *Context:* [task description]
> *The script should be numbered and contains several orders to be called line-by-line in a sequential order.*
> *Use (index) to represent each line.*
> *index starts from 0.*
> *You can use LLM Inference: use LLM("Your Instruction") to find the answer.*
> *Here is one example script for four numbers.* [LWT example]
> *Use (index) to represent the variable you want to replace with previous result.*
> *Use* {(Input)}*,* {(1)}*, ... to represent Input, not allow to directly use numbers.*
> *Use Python indexing to get the element in the list (E.g.,* {(0)}[0]*,* {(0)}[1]*).*
> *Follow the syntax of the example script.*
> *Use your knowledge in the following:*
> [Output of Stage 1]
> *Parse the knowledge;*
> *CAREFULLY CHECK EVERY STEP. MAKE SURE IT HAS THE INPUT FIELD* {(N)} *IT NEEDS TO REFERENCE OUTPUT FROM A PREVIOUS STEP. Make sure it is in the correct syntax. E.g., (from (0)) should be* {(0)}*!*
> *Output only the final script.*

**Task description**

> *context = "You are solving for Game of 24. You need to carefully design an arithmetic expression using ALL the numbers given to you to form 24. Each number must be used exactly once. You*

*may use +, -, ×, and parentheses. Hint: 24 can be obtained by 1×24, 2×12, 3×8, 4×6 or 1+23, 2+22, ..., 12+12. Remember: multiplying by 1 does not change the result. Only use the provided numbers—nothing else."*

**LWT example:** To facilitate better performance, we append the task description to each LWT instruction, providing additional Context and guidance for the LLM.

*(0)=LLM(context+"Given {(Input)}, suggest 12 likely options that combine two numbers using addition, subtraction, or multiplication.")*
*(1)=LLM(context+"Analyze the 12 options from {(0)}. Consider whether the remaining three values can be used to obtain 24.")*
*(2)=LLM(context+"Based on {(1)}, choose the most promising step and Output the state as [expression — remaining values], e.g., [2×3 — 6 4 8].")*
*(3)=LLM("{(2)} Extract the three remaining values from the right-hand side of the bracket.")*
*(4)=LLM(context+"Given {(3)}, suggest 12 possible operations combining two of the three numbers.")*
*(5)=LLM(context+"Analyze the 12 options from {(4)}. Consider whether the remaining value can form 24. Reason step by step.")*
*(6)=LLM(context+"Based on {(5)}, select the best next step.")*
*(7)=LLM(context+"Apply the step from {(6)} and output the result as [expression — two remaining values].")*
*(8)=LLM("{(7)} Extract the two remaining values.")*
*(9)=LLM(context+"How can you make 24 using the two values in {(8)}?")*
*(10)=LLM("{(2)} Extract the expression from the left-hand side of the bracket.")*
*(11)=LLM("{(7)} Extract the expression from the left-hand side of the bracket.")*
*(12)=LLM(context+"Given the steps: Step 1 = {(10)}, Step 2 = {(11)}, Step 3 = {(9)}, combine them to form a full expression that equals 24.")*

**LWT-formatted script:**

*(0)=LLM( "You are solving a 'Game of 24' for five numbers. You must use each number exactly once with +, -, ×, and parentheses to form 24. At each step, you may only combine two of the available numbers (or partial expressions) into a single new partial expression. Given {(Input)}, what are 12 most likely options that combine two of the numbers (using +, -, or ×)?")*
*(1)=LLM("You are solving a 'Game of 24' for five numbers. You must use each number exactly once. We are focusing on the same 12 options we just enumerated: {(0)}. Analyze each option briefly: - Does combining those two numbers in that way keep open a path to 24 with the remaining numbers? - Why or why not? Give me your reasoning for each of the 12 options.")*
*(2)=LLM("{(1)} Take one of the 12 options you find most promising. Perform that operation and produce a partial solution of the form: [arithmetic expression — four remaining values] For example: [2 × 3 5 7 8 — 6 7 8 5] Only Output the new partial expression plus the four leftover numbers in brackets.")*
*(3)=LLM("{(2)} Output the four remaining values on the right side of the bracket.")*
*(4)=LLM("You are solving a 'Game of 24' now for these four numbers: {(3)}. Give me 12 likely ways to combine two of them using +, -, or ×. Note: we will continue trying to reach 24 using exactly these 4 values (plus the partial expression from before).")*
*(5)=LLM("We have the 4 numbers {(3)}. Here are 12 ways to combine two of them: {(4)}. Analyze each of the 12. Which ones might still allow us to reach 24, given the partial expression we already have from {(2)}? Give reasoning.")*
*(6)=LLM("{(5)} Pick the most promising operation among the 12. Perform it and produce a partial solution of the form: [arithmetic expression — three remaining values] Only Output that bracketed state and nothing else.")*
*(7)=LLM("{(6)} Output the three values on the right-hand side of the bracket.")*
*(8)=LLM("We now have 3 numbers or partial expressions: {(7)}. Provide 12 possible ways to combine two of them with +, -, or ×. Our goal remains to reach 24 eventually, using each element exactly once.")*

*(9)=LLM("Given the 3 items {(7)}, here are 12 ways to combine two of them: {(8)}. Analyze these 12 carefully. Which ones might still lead to 24, and why?")*
*(10)=LLM( "{(9)} Pick the best operation from those 12 to continue. Execute it and produce a partial solution of the form: [arithmetic expression — two remaining values] Only Output that bracketed state and nothing else.")*
*(11)=LLM("{(10)} Output the two values on the right-hand side of the bracket.")*
*(12)=LLM("We now have these two items: {(11)}. Show how to combine them (with +, -, or ×) to reach 24. If it's feasible, give the final arithmetic step. If not feasible, explain.")*

*(13)=LLM("{(2)} Output the arithmetic expression on the left side of the bracket.")*
*(14)=LLM("{(6)} Output the arithmetic expression on the left side of the bracket.")*
*(15)=LLM("{(10)} Output the arithmetic expression on the left side of the bracket.")*
*(16)=LLM("We have taken these steps: Step 1: {(13)} Step 2: {(14)} Step 3: {(15)} Step 4: {(12)} Combine them appropriately into a single parenthesized expression (or indicate the correct final expression) that shows how to get 24 from the original five numbers exactly once each.")*

## C   THEORETICAL PROOFS

### C.1   PROOF OF THEOREM 1

**Theorem 1** (Benefit of decomposition). *LLM processes a query of length $L$ correctly with probability*

$$P(L) = \exp(-a\,L^{\gamma}), \; a, \gamma > 0. \tag{3}$$

*If we decompose a length-$L$ task into $K > 1$ equal parts of size $L/K$, then the joint probability of correctly executing the decomposed parts surpasses that of the full-sequence iff $\gamma > 1$.*

*Proof.* According to Levy et al. (2024); Liu et al. (2024), empirical fits on GPT-class models give $1.1 \lesssim \gamma \lesssim 2$ because the error rate compounds superlinearly with sequence length, reflecting limitations in attention stability and context retention over long spans. This compounding arises even when consecutive calls share no hidden state, as is the case with temperature-0 decoding. The success probability of the full-length execution is

$$P_{\text{FULL}} = \exp(-aL^{\gamma}). \tag{4}$$

For the split strategy, each of the $K$ segments has length $L/K$ and succeeds with probability $\exp(-a(L/K)^{\gamma})$. Assuming a bounded recombination overhead (e.g., $O(K^p)$ with $p < \gamma$), the joint probability of success is

$$P_{\text{SPLIT}} = \left[\exp\left(-a\left(\tfrac{L}{K}\right)^{\gamma}\right)\right]^{K} = \exp\left(-aL^{\gamma}K^{1-\gamma}\right). \tag{5}$$

Comparing the two,

$$P_{\text{SPLIT}} > P_{\text{FULL}} \quad \Longleftrightarrow \quad K^{1-\gamma} < 1, \tag{6}$$

which holds for all $K > 1$ iff $\gamma > 1$. Thus, decomposition improves reliability when $\gamma > 1$, even accounting for a modest recombination step. In particular, if the merge step has a sub-quadratic cost (e.g., $O(K^p)$ for $p < \gamma$), the overall probability of success remains higher for the decomposed strategy. Thus, modular execution ($K > 1$) is expected to improve reliability, explaining why XNoT attains significantly better accuracy on 32–64-element inputs where monolithic prompt fails. $\square$

### C.2   PROOF OF LEMMA 1

**Lemma 1** (Recombination overhead). *Suppose a reasoning task of length $L$ is divided into $K$ segments, each of length $L/K$. Then, the total number of LLM calls to solve the task may be proportional to $K^p$ for some overhead exponent $p \geq 0$ due to the need to consolidate each intermediate step into the final answer. For instance, a pairwise merging over the $K$ segments (GoT) results in a balanced tree and incurs $O(K + K\log K)$ steps, i.e., super-linear overhead; a branching strategy*

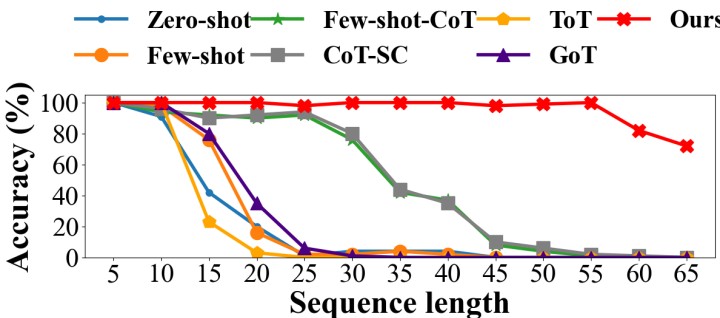

Figure 5: Scalability analysis across varying lengths of addition sequences. XNoT (ours) displays the best consistency performance over extended sequence length.

*repeated $K$ times (ToT) results in $p = 2$ and incurs $O(K + K^2)$ steps, i.e., quadratic overhead.*[8]
*Then, the total number of **reasoning steps** is $O(K + K^p)$, which is at most $O(K^2)$.*

*Proof.* Solving each of the $K$ segments requires a constant number of LLM calls, contributing $O(K)$ total cost. The cost of recombining the intermediate results depends on the structure of the solution strategy. For example, GoT merges results via a balanced tree, requiring $O(K \log K)$ recombination steps. ToT, with repeated branching and pruning, incurs $O(K^2)$ recombination steps. In general, this cost is $O(K^p)$ for some $p \leq 2$. XNoT reduces this overhead by dynamically constructing a dependency graph over elementary steps, reusing intermediate results without rigid merging or redundant branching. Empirically, this yields $p < 2$ in most cases. In the worst case, XNoT can revert to a tree-style merging similar to GoT, yielding $p \leq 2$ since $K \log K < K^2$ for $K \geq 2$. Therefore, the total number of reasoning steps is $O(K) + O(K^p) = O(K + K^p)$, with $p \leq 2$ for all $K \geq 2$. □

### C.3 PROOF OF LEMMA 2

**Lemma 2** (Decay vs. overhead). *Under equation 3 and overhead exponent $p$, an LLM answers a decomposed task correctly with probability $P_{\text{SPLIT}} = \exp[-aL^\gamma K^{p-\gamma}]$.*

*Proof.* From Eq. equation 2, the probability of correctly processing a length-$L$ input is $\exp(-aL^\gamma)$. Decomposing a length-$L$ task into $K$ segments of size $L/K$ produces $K$ calls, each with success probability $\exp(-a(L/K)^\gamma)$. By equation 5, the joint success probability across all $K$ segments is $\exp\left[-aL^\gamma K^{1-\gamma}\right]$. Next, by Lemma 1 we assume that the recombination requires $O(K^p)$ calls, each also operating on inputs of length $L/K$. Each merge call contributes a failure factor $\exp(-a(L/K)^\gamma)$, so the final total recombination success rate is $\exp\left[-aL^\gamma K^{p-\gamma}\right]$, as claimed. □

## D ADDITIONAL EXPERIMENT AND DETAILS

### D.1 ADDITIONAL EXPERIMENTS

#### D.1.1 LENGTH SCALABILITY EXPERIMENT

XNoT effectively achieves sequence-length generalization, retaining stable performance as problem sizes grow. In Table 2, baselines often collapse to near $0\%$ accuracy on larger inputs (e.g., 64-element sorting, 128-element set intersection, or 32-number arithmetic), whereas XNoT still achieves $27\%$, $20\%$, and $10\%$, respectively. In addition, Fig. 5 shows detailed scalability tests for computing pure addition sequences ranging from 5 to 65 numbers, confirming XNoT's robustness in scaling to longer sequences. While all methods start near $100\%$ accuracy at length 5, most degrade

---

[8]While ToT processes the full query at each step, it explores partial solutions, requiring $K$ repetitions with $K$-way branching. Pruning to one branch per step still results in $O(K^2)$ evaluations, yielding quadratic overhead.

Table 7: Ablating *Knowledge Extraction Prompt*.

| | ① | ② | ③ | ①+② | ①+③ | ②+③ | XNoT |
|---|---|---|---|---|---|---|---|
| keyword counting-5 | 59% | 94% | 67% | 94% | 88% | 98% | 100% |
| sorting-16 | 76% | 90% | 84% | 92% | 90% | 96% | 100% |
| set intersection-32 | 68% | 91% | 75% | 93% | 92% | 97% | 100% |
| arithmetic-8 | 88% | 92% | 94% | 95% | 99% | 98% | 100% |
| large-digit-8 | 51% | 91% | 60% | 94% | 85% | 98% | 100% |

Table 8: Ablating *LWT Compilation Prompt*.

| | ④ | ⑤ | ⑥ | ④+⑤ | ④+⑥ | ⑤+⑥ | XNoT |
|---|---|---|---|---|---|---|---|
| keyword-5 | 0% | 0% | 92% | 0% | 96% | 97% | 100% |
| sorting-16 | 0% | 0% | 88% | 0% | 95% | 92% | 100% |
| set intersection-32 | 0% | 0% | 91% | 0% | 97% | 99% | 100% |
| arithmetic-8 | 0% | 0% | 94% | 0% | 96% | 98% | 100% |
| large-digit-8 | 0% | 0% | 92% | 0% | 93% | 97% | 100% |

quickly between lengths 10 and 20, with CoT and CoT-SC dropping to 0% by 30–45. In contrast, XNoT remains consistent until beyond length 55, highlighting its stronger scalability.

### D.1.2 ABLATION STUDY

Tables 7 and 8 present the ablation study results, evaluating the contribution of individual components within XNoT. In Table 7, ①, ②, and ③ correspond to the task description $C$, the node simplicity prompt, and the edge simplicity prompt within the *Knowledge Extraction Prompt*, respectively. The results indicate that while the node simplicity prompt alone achieves decent accuracy (greater than 90%), the complete XNoT significantly outperforms all ablation versions, emphasizing the importance of all three components.

In Table 8, ④, ⑤, and ⑥ correspond to the task description $C$, the compilation instructions, and the LWT example $\mathcal{E}$ within the *LWT Compilation Prompt*. The results in Table 8 show a more drastic difference, with the LWT example $\mathcal{E}$ proving to be the most critical component. Nevertheless, the other two components still contribute positively to achieving 100% accuracy obtained by the full XNoT. These findings collectively demonstrate that XNoT's design, incorporating all components, is essential for optimal performance.

### D.1.3 QUALITATIVE ANALYSIS

Table 9 presents a qualitative study of the trade-off between computational cost and workflow complexity. Decomposition into finer steps increases the number of LLM calls and therefore runtime overhead, whereas coarser steps reduce runtime but compromise execution reliability.

Importantly, the structural complexity of the workflow itself does not add runtime overhead during Stage 3. All message passing and input substitutions are resolved externally prior to LLM invocation. Once the LWT script is compiled, each instruction executes independently, and the dominant cost is determined solely by the number of LLM calls, governed by the granularity of decomposition.

The results in Table 9 highlight this trade-off on the addition task. For 1-by-1 decomposition, accuracy remains at 100% despite longer runtime. In contrast, 2-by-2 and 3-by-3 decompositions reduce the number of steps and total runtime by nearly half and two-thirds, respectively, but exhibit severe degradation in accuracy. The instability arises because multi-number additions per step are less intuitive for the model, even when explicit contextual prompts are provided.

These findings demonstrate that while coarse decomposition improves efficiency, the correctness loss is substantial. We therefore conclude that XNoT benefits most from **atomic decomposition**, where each step is simple, explicit, and reliably executed by the model.

Table 9: Runtime and accuracy under different decomposition granularities on the addition task.

| Decomposition | # Steps | Avg. Latency Per Step (s) | Avg. Total Runtime (s) | Worst-case Runtime (s) | Accuracy |
|---|---|---|---|---|---|
| 1-by-1 | 34 | 0.55 | $18.19 \pm 0.87$ | 19.28 | 100% |
| 2-by-2 | 19 | 0.56 | $9.49 \pm 0.60$ | 10.58 | 23% |
| 3-by-3 | 14 | 0.54 | $6.65 \pm 0.59$ | 7.48 | 0% |

### D.1.4    5 NUMBER GAME OF 24

We demonstrate XNoT's intelligence amplification capabilities on the Game of 24, a mathematical puzzle requiring players to combine four numbers with arithmetic operations to yield 24. A key limitation of existing algorithmic prompting methods (such as ToT and GoT) is their inability to directly generalize from 4-number to 5-number variants without substantial manual redesign, as they rely on hard-coded, instance-specific solution structures.

We evaluate whether XNoT can autonomously scale from solving the 4-number task to the more complex 5-number variant with intelligence amplification. We begin with a LWT example for the original 4-number setting. Following standard solution strategies (AlgoMonster Contributors, n.d.), the LWT employs a branch-and-prune approach, prompting the LLM to propose candidate operations between number pairs, assess the resulting expressions, and iteratively choose the most promising path. Notably, the versatility of the LWT format enables seamless integration of this algorithmic structure without altering the XNoT workflow. This illustrates that XNoT not only supports algorithmic prompting techniques but also generalizes to more complex variants with minimal human intervention. We then apply XNoT to generate a LWT-formatted script for the 5-number Game of 24.[9]

To ensure solvability, we construct the five-number instances by appending 1 to existing four-number examples. The resulting 5-number LWT-formatted scripts achieve a 24% success rate without manual algorithm redesign, whereas the baseline methods (Sel et al., 2024; Yao et al., 2024) effectively achieve 0%, as they are tailored to the 4-number variant and do not provide solutions for the 5-number variant. This demonstrates XNoT's intelligence amplification capabilities, allowing it to harness LLM capabilities and adapt reasoning structures encoded by LWT-formatted scripts autonomously to more complex problem variants.

### D.1.5    GSM8K

GSM8K consists of instance-specific grade-school math problems that do not naturally showcase corpus-level reuse or scalable *intelligence amplification*. Consequently, this benchmark is not ideal for evaluating XNoT's amplification capabilities. Prior work has likewise treated GSM8K as comparatively straightforward for well-designed prompting methods (Yao et al., 2024).

Nevertheless, we evaluate an *abridged* version of XNoT on GSM8K. We employ a constant LWT script (shared across all problems; shown below) and a zero-shot breadth-first search strategy inspired by Tree of Thoughts (ToT): generate five independent candidate solutions and select the best via an explicit aggregation operator. The key difference from ToT is that LWT specifies the aggregation step declaratively inside the prompt, enabling precise, auditable selection without external orchestration.

Experimental results are reported in Table 10. Accuracy for *Input–Output*, *Chain-of-Thought*, and *Tree-of-Thought* baselines is taken from Yao et al. (2024). Our evaluation protocol and model choice (GPT-4) follow the same setup as Yao et al. (2024). As shown in Table 10, despite using a comparable search strategy, XNoT attains higher accuracy, which we attribute to LWT's prompt-native aggregation that improves consistency in candidate selection.

> *(0)=LLM("You are solving a math problem. Given the question "'{(input)}'", list calculation process step-by-step then output the final answer.")*

---

[9]Details of XNoT's prompt design for the Game of 24 are provided in Appendix B.3.

Table 10: GSM8K experiment results

| Method | Input Output | Chain of Thought | Tree of Thought | **XNoT** |
|--------|--------------|------------------|-----------------|----------|
| Accuracy | 51% | 86% | 90% | **94%** |

*(1)=LLM("You are solving a math problem. Given the question "'{(input)}"', list calculation process step-by-step then output the final answer.")*
*(2)=LLM("You are solving a math problem. Given the question "'{(input)}"', list calculation process step-by-step then output the final answer.")*
*(3)=LLM("You are solving a math problem. Given the question "'{(input)}"', list calculation process step-by-step then output the final answer.")*
*(4)=LLM("You are solving a math problem. Given the question "'{(input)}"', list calculation process step-by-step then output the final answer.")*
*(5)=LLM("You are solving a math problem. The problem is "'{(input)}"'. You need to carefully check the following five calculation processes and answers, then choose the best final answer: '(0)', '(1)', '(2)', '(3)', '(4)'. In the end, extract the numerical value of the final answer you choose and print the value without anything else.")*

### D.1.6 HEALTHCARE TRIAGE

Healthcare triage involves structured decision-making processes where practitioners follow established protocols to assess patient severity and determine appropriate care pathways. To evaluate XNoT's ability to execute sequential, rule-based reasoning, we construct four synthetic triage workflows.

Listing 1: Vitals-First with Comorbidity Overlay

```
"workflow_name": "Vitals-First with Comorbidity Overlay",
"logic_flow": "If (O2 saturation < 92 OR systolic BP < 90 OR HR > 130):
    -> severity = critical
 Else If (O2 < 95 OR Temp > 101°F OR RR > 24):
    -> severity = moderate
 Else:
    -> severity = mild

 If (>=2 chronic conditions OR age >= 70):
    -> risk = high
 Else:
    -> risk = standard

 If (severity = critical):
    -> recommend ER referral
 Else If (severity = moderate AND risk = high):
    -> recommend urgent clinical evaluation
 Else If (severity = moderate AND risk = standard):
    -> recommend outpatient evaluation
 Else:
    -> recommend home care"
```

Listing 2: Symptom Cluster Then Escalation

```
"workflow_name": "Symptom Cluster Then Escalation",
"logic_flow": "If (chest pain OR shortness of breath OR confusion):
    -> red_flag = true
 Else If (fever AND cough AND fatigue):
    -> viral_cluster = likely
 Else:
    -> viral_cluster = unlikely

 If (red_flag):
```

```
    -> recommend ER referral
Else If (viral_cluster AND symptoms < 3 days):
    -> recommend home care
Else If (viral_cluster AND symptoms >= 3 days):
    -> recommend outpatient evaluation
Else:
    -> recommend clinical judgment follow-up"
```

Listing 3: Duration-Weighted Symptom Score

```
"workflow_name": "Duration-Weighted Symptom Score",
"logic_flow": "symptom_score = 0
If (cough): +1
If (fever): +1
If (shortness of breath): +2
If (symptom duration > 7 days): +1

If (symptom_score >= 4):
    -> severity = high
Else If (symptom_score >= 2):
    -> severity = moderate
Else:
    -> severity = low

If (severity = high):
    -> recommend urgent clinical evaluation
Else If (severity = moderate):
    -> recommend outpatient evaluation
Else:
    -> recommend home care"
```

Listing 4: Immunosuppression Priority Pathway

```
"workflow_name": "Immunosuppression Priority Pathway",
"logic_flow": "If (on chemotherapy OR HIV+ with CD4 < 200 OR chronic
    steroid use):
    -> immunosuppressed = true
Else:
    -> immunosuppressed = false

If (any sign of infection: fever, chills, or cough):
    -> infection_suspected = true
Else:
    -> infection_suspected = false

If (immunosuppressed AND infection_suspected):
    -> recommend urgent clinical evaluation
Else If (infection_suspected):
    -> recommend outpatient evaluation
Else:
    -> recommend home care"
```

We then generate ground-truth input–output pairs for evaluation:

Listing 5: Ground-truth examples

```
"A 34-year-old patient presents with dry cough for 1 day, rash for 2 days
    , diarrhea for 1 day, fatigue for 2 days. Oxygen saturation is 97%,
    and body temperature is 98.6°F. Medical history includes CHF, asthma.
     Recent hospital stay.",
Workflow: Symptom Cluster Then Escalation,
Answer: Clinical judgment follow-up
```

```
"A 26-year-old patient presents with headache for 7 days, rash for 6 days
    , dry cough for 4 days, chest pain for 9 days. Oxygen saturation is
    90%, and body temperature is 98.6°F. Medical history includes none.
    Recent travel to
outbreak area.",
Workflow: Duration-Weighted Symptom Score,
Answer: Outpatient evaluation

"A 75-year-old patient presents with chest pain for 7 days, fatigue for 4
days, dry cough for 8 days. Oxygen saturation is 94%, and body
    temperature
is 100.4°F. Medical history includes none. No known exposure.",
Workflow: Immunosuppression Priority Pathway,
Answer: Outpatient evaluation
```

We leverage a LWT example for *Vitals-First with Comorbidity Overlay* and allow XNoT to automatically generate equivalent LWT scripts for the other workflows. In particular, the LWT example for *Vitals-First with Comorbidity Overlay* is as follows.

*(0)=LLM("Extract structured fields from the input. Return a Python dictionary with the following keys: 'oxygen' (int), 'temperature' (float), 'age' (int), 'comorbidities' (list of strings). Input: (input)")*
*(1)=LLM("Is oxygen saturation less than 92? Based on (0). Output 'yes' or 'no'.")*
*(2)=LLM("Is oxygen saturation less than 95? Based on (0). Output 'yes' or 'no'.")*
*(3)=LLM("Is temperature greater than 101F? Based on (0). Output 'yes' or 'no'.")*
*(4)=LLM("Let Q1 indicate whether oxygen saturation is less than 92. If Q1 = 'yes', then severity = critical. Given Q1 = (1), does this branch apply? Output only one of: applies, does not apply.")*
*(5)=LLM("If oxygen saturation < 95 OR temperature > 101F, then severity = moderate. This applies only if Q1 is 'no' and (Q2 is 'yes' OR Q3 is 'yes'). Given Q1=(1), Q2=(2), Q3=(3), does this branch apply? Output only one of: applies, does not apply.")*
*(6)=LLM("If none of the previous conditions apply, then severity = mild. This applies only if Q1 is 'no', Q2 is 'no', and Q3 is 'no'. Given Q1=(1), Q2=(2), Q3=(3), does this branch apply? Output only one of: applies, does not apply.")*
*(7)=LLM("Only one severity level should apply. Severity branch outcomes: - critical: (4) - moderate: (5) - mild: (6) Return the first severity level that applies.")*
*(8)=LLM("Is the patient's age greater than or equal to 70? Use (0). Output 'yes' or 'no'.")*
*(9)=LLM("(0). Look at the ['comorbidities'] list and count how many items are in it. If the count is greater than or equal to 2, output 'yes'. Otherwise, output 'no'.")*
*(10)=LLM("Determine risk level using previous answers: If Q1 is 'yes' OR Q2 is 'yes' → risk = high. Else → risk = standard. Use Q1=(8), Q2=(9). Output only one of: high, standard.")*
*(11)=LLM("Rephrase the decision logic step as follows: If severity is critical → ER referral. Given severity=(7) and risk=(10), Does this branch applies (applies, does not apply)?")*
*(12)=LLM("Rephrase the decision logic step as follows: If severity is moderate and risk is high → Urgent clinical evaluation. Given severity=(7) and risk=(10), Does this branch applies (applies, does not apply)?")*
*(13)=LLM("Rephrase the decision logic step as follows: If severity is moderate and risk is standard → Outpatient evaluation. Given severity=(7) and risk=(10), does this branch applies (applies, does not apply)?")*
*(14)=LLM("Rephrase the decision logic step as follows: If severity is mild → Home care. Given severity=(7) and risk=(10), does this branch applies (applies, does not apply)?")*
*(15)=LLM("The result of each branch is: ER referral (11). Urgent clinical evaluation (12). Outpatient evaluation (13). Home care (14). Output the one that is applicable")*

The "Symptom Cluster Then Escalation" workflow generated automatically by XNoT is given as an example as follows.

*(0)=LLM("Extract structured fields from the input. Return a Python dictionary with the following keys: 'age' (int), 'symptoms' (list of dicts with 'name' and 'duration_days'), 'oxygen' (int),*

Table 11: Healthcare triage experiment results.

| Method | CoT | ZeroCoT | **XNoT** |
|--------|-----|---------|----------|
| Accuracy | 21% | 40% | **62**% |

> 'temperature' (float), 'comorbidities' (list of strings), 'recent_hospital_stay' (bool). Input: (input)")
> (1)=LLM("Does the patient have any of the following symptoms: chest pain, shortness of breath, or confusion? Use the 'symptoms' list from (0). Output 'yes' or 'no'.")
> (2)=LLM("If the patient does not have chest pain, shortness of breath, or confusion, then red_flag = false. Given (1), is red_flag false? Output 'yes' or 'no'.")
> (3)=LLM("Does the patient have all of the following symptoms: fever, cough, and fatigue? Use the 'symptoms' list from (0). Output 'yes' or 'no'.")
> (4)=LLM("If the patient does not have all of fever, cough, and fatigue, then viral_cluster = unlikely. Given (3), is viral_cluster unlikely? Output 'yes' or 'no'.")
> (5)=LLM("If red_flag is true, recommend ER referral. Given red_flag=(1), does this branch apply? Output only one of: applies, does not apply.")
> (6)=LLM("If viral_cluster is likely and all symptom durations are less than 3 days, recommend home care. Use viral_cluster=(3) and the 'symptoms' list from (0). Does this branch apply? Output only one of: applies, does not apply.")
> (7)=LLM("If viral_cluster is likely and any symptom duration is greater than or equal to 3 days, recommend outpatient evaluation. Use viral_cluster=(3) and the 'symptoms' list from (0). Does this branch apply? Output only one of: applies, does not apply.")
> (8)=LLM("If none of the previous branches apply, recommend clinical judgment follow-up. Given ER referral=(5), home care=(6), outpatient evaluation=(7), does this branch apply? Output only one of: applies, does not apply.")
> (9)=LLM("The result of each branch is: ER referral (5). Home care (6). Outpatient evaluation (7). Clinical judgment follow-up (8). Output the one that is applicable.")

The modular decomposition approach of XNoT contrasts with CoT and ZeroCoT, which often fail to disentangle composite conditions or preserve sequential dependencies. As a result, they misclassify intermediate variables (e.g., severity, risk), leading to incorrect final recommendations. For instance, given input:

Listing 6: Failure case of CoT

```
"A 51-year-old patient presents with loss of smell for 1 day,
chest pain for 7 days, headache for 7 days. Oxygen saturation is 90%,
and body temperature is 102.0°F. Medical history includes HIV.
Recent travel to outbreak area."
Workflow: Vitals-First with Comorbidity Overlay
```

CoT incorrectly reasons:

```
Check severity:
- O2 saturation < 92
- Temp > 101
-> severity = moderate
```

Despite the oxygen saturation of 90% clearly satisfying the *critical* threshold.

In contrast, XNoT avoids such errors by explicitly verifying each condition through fine-grained, step-wise evaluation. This structured decomposition ensures that intermediate variables such as severity and risk are assessed consistently, reducing the likelihood of logical oversights or shortcut reasoning. As a result, XNoT substantially outperforms baselines such as ZeroCoT and CoT, which often fail to disentangle composite conditions and thus misclassify intermediate states.

Beyond handling a single triage workflow, XNoT generalizes this explicit approach across diverse healthcare triage protocols, automatically adapting to alternative branching logic without requiring

Table 12: More experiment results of arithmetic calculation with pure addition as well as "addition and multiplication."

| method | addition | | | addition and multiplication | | |
|--------|------|------|------|------|------|------|
| | 8 | 16 | 32 | 8 | 16 | 32 |
| Few Shot | 94% | 76% | 4% | 24% | 0% | 0% |
| CoT | 98% | 92% | 64% | 68% | 35% | 2% |
| CoT-SC | 100% | 93% | 62% | 69% | 33% | 4% |
| ToT | 100% | 23% | 0% | 52% | 19% | 6% |
| GoT | 100% | 80% | 1% | 28% | 1% | 0% |
| **XNoT** | 100% | 100% | 100% | 98% | 56% | 20% |

task-specific redesign. This demonstrates both robustness and scalability in domains where rule-based reasoning is critical. Table 11 summarizes the comparative accuracy results.

### D.1.7 EXPERIMENT RESULTS ON SEQUENCE OF ONLY ADDITION AND MULTIPLICATION

In the primary arithmetic calculation task, we tested all four arithmetic operations: addition, multiplication, subtraction, and division. Here, to evaluate the discrepancy of baselines, we present additional experimental results focused on "pure addition" and calculations involving only "addition and multiplication." As shown in Table 12, XNoT consistently outperforms all the other baselines in these scenarios as well. In contrast, existing baselines degrade significantly as sequence length increases, especially in the presence of mixed operations. This experiment highlights that while some baselines handle simple addition reasonably well, they struggle with scalability and compositional reasoning, even under restricted operator sets. By isolating these operations, we expose the limitations of prior prompt schemes and demonstrate XNoT's robustness in handling both long sequences and compositional arithmetic.

### D.2 EXPERIMENT SETUP AND CONFIGURATIONS

In the following, we provide experiment details, including the task query example and manual prompt designs utilized for the baseline methods. Note that due to budget constraints, we first evaluate all approaches on GPT-3.5-turbo and then compare them in GPT-4o. Consistent with findings reported in GoT (Besta et al., 2024a), we also observe that open-source models such as LLaMA (Touvron et al., 2023) generally perform worse than GPT-3.5 and are slower to run, especially when applied to larger divisions, which makes them impractical under limited computational resources. We therefore follow prior works (Zhou et al., 2023; Wang et al., 2023; Yao et al., 2024; Sel et al., 2024) in using high-performance API-based language models, which remain the standard choice for evaluating advanced prompting strategies under practical resource constraints.

### D.2.1 INPUT QUERY EXAMPLE

In this section, we will provide one input query example for each task.

**Keyword counting** The input is an article that contains 14 to 20 sentences. We only list one of them in the following:

> *John, an avid traveler from Canada, had spent his summer exploring the heart of Australia, specifically, the Outback. The vast, arid landscapes of Australia presented a stark contrast to the snow-filled winters of his home in Canada, and he reveled in the difference. He then shared stories of his trip to Brazil, where he fell in love with the vibrant rhythms and the people's warm hospitality. Indeed, Brazil left such a strong impression on him that he visited the country again, this time to explore the dense Amazon rainforest. As John recounted his travels, his friend Sarah, a history buff from the United Kingdom, couldn't help but gush about her trips to Italy and Greece. She explained how she had spent weeks soaking up the culture, history, and mythology of Italy and Greece. Intrigued by Sarah's stories, John revealed his fascination for Northern*

*countries, particularly Norway and Sweden. He cherished his memories of hiking through the scenic landscapes of Norway and the breathtaking fjords of Sweden. Sarah, not to be outdone, discussed her recent visit to Mexico and Cuba. Highlighting the unique colonial architecture of Mexico and the vibrant music scene in Cuba, Sarah couldn't conceal her wanderlust. She ended the conversation by expressing her desire to visit South Korea and Japan. She was particularly interested in the modern cities and ancient temples of South Korea, as well as the unique blend of tradition and technology in Japan. As they parted ways, both agreed to continue exploring and understanding the world, one country at a time.*

**Sorting**    In the sorting task, the input is an unsorted array containing duplicate numbers.

The following example is length 16:

*[1, 2, 6, 1, 1, 6, 0, 3, 7, 4, 5, 2, 9, 2, 1, 5]*

The following example is length 32:

*[0, 0, 5, 9, 0, 7, 9, 9, 1, 2, 6, 1, 1, 9, 0, 1, 3, 5, 2, 3, 5, 6, 0, 2, 7, 4, 6, 2, 9, 7, 9, 5]*

The following example is length 64:

*[6, 3, 6, 5, 1, 2, 4, 3, 8, 0, 7, 8, 6, 4, 9, 5, 2, 4, 8, 4, 4, 4, 5, 6, 8, 4, 7, 7, 8, 9, 4, 9, 5, 4, 8, 4, 0, 5, 6, 9, 1, 2, 3, 6, 2, 0, 8, 1, 0, 7, 1, 2, 0, 7, 6, 9, 9, 9, 5, 6, 8, 3, 9, 0]*

**Set operation**    The input contains two sets without duplicate numbers.

The following example is length 32:

*Set1: [11, 60, 1, 49, 21, 33, 14, 56, 54, 15, 23, 40, 45, 22, 7, 28, 20, 46, 51, 6, 34, 37, 3, 50, 17, 8, 25, 0, 35, 47, 18, 19]*
*Set2: [31, 11, 4, 63, 38, 58, 59, 24, 61, 14, 32, 39, 27, 46, 48, 19, 52, 57, 50, 56, 3, 2, 53, 29, 5, 37, 62, 41, 36, 12, 49, 16]*

The following example is length 64:

*Set1: [42, 73, 86, 39, 85, 77, 69, 59, 43, 127, 121, 88, 109, 53, 70, 66, 25, 51, 34, 78, 45, 11, 40, 99, 68, 47, 49, 41, 101, 31, 24, 84, 36, 29, 118, 75, 3, 27, 30, 80, 125, 8, 37, 46, 90, 21, 60, 83, 19, 6, 95, 117, 87, 18, 100, 13, 22, 10, 110, 102, 35, 81, 17, 63]*
*Set2: [34, 49, 116, 106, 112, 23, 5, 80, 18, 62, 90, 54, 32, 103, 37, 43, 9, 25, 92, 16, 111, 79, 64, 91, 107, 58, 72, 94, 7, 60, 33, 14, 19, 104, 28, 74, 96, 76, 38, 52, 114, 50, 17, 0, 3, 100, 69, 98, 2, 1, 99, 12, 95, 97, 123, 4, 126, 124, 82, 27, 67, 57, 115, 46]*

The following example is length 128:

*Set1: [132, 75, 157, 25, 199, 202, 147, 109, 221, 110, 220, 251, 213, 11, 224, 101, 200, 170, 155, 71, 119, 122, 39, 1, 29, 113, 189, 212, 10, 219, 49, 28, 151, 40, 103, 8, 145, 214, 114, 91, 175, 107, 152, 163, 148, 246, 176, 181, 18, 106, 74, 115, 144, 0, 205, 121, 46, 234, 142, 223, 228, 162, 96, 97, 130, 156, 172, 241, 33, 186, 137, 150, 65, 161, 226, 116, 111, 12, 146, 38, 167, 4, 108, 169, 61, 93, 190, 252, 22, 31, 3, 9, 13, 35, 23, 141, 129, 198, 85, 84, 62, 158, 201, 67, 117, 59, 41, 191, 56, 90, 51, 227, 143, 83, 184, 174, 125, 98, 232, 238, 57, 225, 54, 179, 177, 237, 37, 95]*
*Set2: [27, 162, 187, 254, 128, 227, 2, 165, 143, 109, 140, 46, 160, 26, 139, 171, 42, 199, 207, 30, 205, 117, 213, 48, 40, 212, 185, 196, 197, 94, 136, 35, 229, 193, 36, 7, 15, 43, 4, 203, 142, 144, 49, 31, 240, 124, 116, 69, 37, 250, 95, 105, 103, 168, 126, 64, 73, 206, 24, 157, 135, 118, 34, 134, 45, 62, 153, 5, 47, 239, 216, 222, 80, 231, 102, 21, 57, 215, 149, 141, 236, 32, 188, 204, 194, 23, 233, 83, 154, 210, 159, 70, 202, 253, 20, 71, 166, 242, 221, 228, 78, 230, 29, 145, 147, 81, 104, 235, 66, 100, 131, 132, 244, 195, 68, 72, 53, 182, 79, 248, 3, 82, 211, 173, 180, 17, 77, 51]*

**Arithmetic sequence calculation**    The input contains an arithmetic sequence.

The following example is length 8:

> *6+4+3+3\*3+2+4+2*

The following example is length 16:

> *2/9-3-4+6+4-9+8+8-4\*5-7+2/1+6+7*

The following example is length 32:

> *8-2/2/9+9\*1/7/3\*4+2/5-9+4\*8+5+8+9+5+5-2+7/2-2+6-8+7+6+5+1+6\*3+1*

**Large digit addition**    The input is the addition of two large-digit numbers.

The following example is length 8:

> *57247728+67594862*

The following example is length 16:

> *5465458164972518+8654164596886757*

The following example is length 32:

> *5984282913361747342716688425297z+2487337637186337169898274489z145*

### D.2.2    MANUAL PROMPTS DESIGN FOR BASELINE PROMPT SCHEMES

We provide manual designs for tasks not covered in baseline prompt schemes. In particular, the following template is used for **keyword counting**, **sorting**, and **set intersection** for CoT, ToT, and GoT based on the open source code provided by GoT (Besta et al., 2024a). The following details prompt design for the **arithmetic calculation** task. Please view the detailed prompt for **Yelp review comprehension** and **large digit addition** in our supplementary files.

**Few shot arithmetic example for Chain-of-thoughts**    We provide a step-by-step calculation example for the CoT prompt scheme as follows. In particular, we present an example of a problem size that is the same as the target task. It is worth noting that XNoT leverages the same example of *shorter* problem size across different problem sizes, demonstrating its effectiveness in reducing human labor.

> *<Example>*
> *Input: 3+5+6+2+4+5\*3+2*
> *Answer: 3+5=8, 8+6=14, 14+2=16, 16+4=20, 5\*3=15, 20+15=35, 25+2=37.*
> *The final answer is 37.*

> *<Example>*
> *Input: 7+4+1\*6+7+3+7+2+2\*7+3+3\*6+2+5+4*
> *Answer: 7+4=11, 1\*6=6, 11+6=17, 17+7=24, 24+3=27, 27+7=24, 34+2=36, 2\*7=14,*
> *36+14=50, 50+3=53, 3\*6=18, 53+18=71, 71+2=73, 73+5=78, 78+4=82.*
> *The final answer is 82.*

> *<Example>*
> *Input: 7+6+2+7+3+6+5\*2+4+2+4+7+2+4+3\*3*
> *+3+5+4+7+6+4+6+7+6+5\*2\*7+7+3+7+7*
> *Answer: 7+6=13, 13+2=15, 15+7=22, 22+3=25, 25+6=31, 52=10, 31+10=41, 41+4=45,*
> *45+2=47, 47+4=51, 51+7=58, 58+2=60, 60+4=64, 33=9, 64+9=73, 73+3=76, 76+5=81,*

> *81+4=85, 85+7=92, 92+6=98, 98+4=102, 102+6=108, 108+7=115, 115+6=121, 527=70, 121+70=191, 191+7=198, 198+3=201, 201+7=208, 208+7=215.*
> *The final answer is 215.*

**Manual module preparation for ToT**  The detailed mechanism for ToT and GoT is introduced as follows. ToT first repeatedly use a *Calculation Module* to perform initial calculation attempts. Then, it selects the best attempt by prompting the LLM to score its own attempt. Afterwards, ToT leverages the *Improve Module* to make improvements based on the current and previous results (i.e., a node and its parent node in the tree of thoughts).

Following (Besta et al., 2024a), the *Calculation Module* includes a few-shot examples for the targeted problem sizes and is prepared as follows.

> *<Instruction> Calculate the given sequence. Output only the number, no additional text.*
> *<Example>*
> *Input: 3+5+6+2+4+5*3+2*
> *Output: 37*
> *Input: 7+4+7*6+7+3+7+2+7*7+3+3*6+2+5+4*
> *Output: 153*
> *Input: 7+6+2+7+3+6+5*2+4+2+4+7+2+4+3*3+3+*
> *5+4+7+6+4+6+7+6+5*2*7+7+3+7+7*
> *Output: 215*

The *Improve Module* is prepared as follows.

> *<Instruction> There are some errors in the following calculation sequence. Find the errors in it and correct them.*
> *<Approach>*
> *To fix the incorrect answer, follow these steps:*
> *1. Check all numbers in the sequence one by one.*
> *2. Attention to the symbol error using.*
> *<Example>*
> *Input: 3+5+6+2+4+5*3+2*
> *Incorrectly Answer: 39*
> *Reason: Add 2 one more time*
> *Output: 37*
> *Input: 7+4+7*6+7+3+7+2+7*7+3+3*6+2+5+4*
> *Incorrectly Answer: 149*
> *Reason: Forgot to add 4, the last number in the sequence*
> *Output: 153*
> *Input: 7+6+2+7+3+6+5*2+4+2+4+7+2+4+3*3*
> *+3+5+4+7+6+4+6+7+6+5*2*7+7+3+7+7*
> *Incorrectly Answer: 202*
> *Reason: The incorrect addition of the first two numbers, remember to add.*
> *Output: 215*

Specifically, we record the mistakes that LLMs make under the ToT scheme, and iteratively append the corrections into the *Improve Module*.

**Manual module preparation for GoT**  GoT follows ToT to use the same *Calculation Module* and *Improvement Module*. Besides these, it additionally leverages the *Split Module* and the *Merge Module*. Concretely, GoT first splits the input task into several equal-sized chunks, then applies the initial calculation and repeated improvements to each chunk, and finally merges the results from all chunks.

The following is the *Split Module*.

> *<Instruction> Split the following sequence of 8 numbers into two sequence of 4 numbers each, the first sequence should contain the first four numbers and the second sequence the second 4 numbers.*
> *Only output the final 2 sequences in the following format without any additional text or thoughts!:*
> *"sequence 1": 3+4+5\*1+,*
> *"sequence 2": 5+2+3\*4*
> *<Example>*
> *Input: 3+5+6+2+4+5\*3+2*
> *Output:*
> *"sequence 1": 3+5+6+2+,*
> *"sequence 2": 4+5\*3+2*
> *Input: 7+4+7\*6+7+3+7+2+7\*7+3+3\*6+2+5+4*
> *Output:*
> *"sequence 1": 7+4+7\*6+7+3+7+2+,*
> *"sequence 2": 7\*7+3+3\*6+2+5+4*

The following is the *Merge Module*.

> *<Instruction> Merge the following 2 final answers. Only output the final number without any additional text or thoughts!*
> *<Approach>*
> *To merge the two numbers, follow these steps:*
> *1. Calculate the answer of 2 numbers*
> *<Example>*
> *Input:*
> *"sequence 1": 14,*
> *"sequence 2": 28*
> *Output: 42*

It is worth noting that the split-then-merge structure is antithetical to the arithmetic calculation task, as the operation in the middle of the sequence is not necessarily addition or subtraction. Nevertheless, we designed the addition operation for the merge module and further evaluated all prompt schemes with pure addition sequences. As shown in Fig. 5 and Table 12, GoT finds better performance compared to arithmetic calculations involving all four operations. However, it still fails in comparison to XNoT as well as CoT-based schemes.

## D.3 USE OF LARGE LANGUAGE MODELS (LLMS)

We used OpenAI ChatGPT (GPT-5 Thinking; Aug–Sep 2025) only for (i) light writing polish (grammar, clarity, and style on author-written text) and (ii) literature search via keyword brainstorming. We did not use LLMs to create technical claims, design methods, run experiments, or write sections without human verification. All edits and references were checked by the authors; no confidential data were shared; interactions were inference-only. On the submission form we selected "Yes, to aid or polish writing" and "Yes, for retrieval and discovery."

