# OpenReview forum: "Executable Networks of Thought: Scaling Reasoning with LLM Workflow Template"
_ICLR.cc/2026/Conference — ICLR 2026 Conference Withdrawn Submission_

### Official Review · Reviewer_7Cp5 · 2025-10-29

**Soundness:** 2
**Presentation:** 3
**Contribution:** 2
**Rating:** 4
**Confidence:** 3

**Summary:**

This paper presents a prompting method named Executable Network of Thoughts (XNoT). During the prompting process, the LLM is instructed to generate a network of thought dependencies among sequential elementary steps. The authors mainly test their prompting method on GPT-3.5-Turbo models on five synthetic datasets, and the proposed method outperforms the existing prompting methods.

**Strengths:**

- The paper is clearly written and easy to follow.
- I generally like the idea of making the LLM generate the so-called LWT (LLM workflow template) script, where the dependencies are explicitly listed out during the generation process. It seems to me that the core idea is to reduce the burden of LLMs on attending to the dependency issues (which is handled by the execution of the script itself) in its reasoning process.

**Weaknesses:**

- Like what I mentioned, the shining point for the method seems to be casting the dependency consideration to the script execution process. In that sense, is this method merely a program of thought with some kinds of execution? And how does this method compare to program of thoughts [1]?

- The tasks presented in the paper are rather synthetic, any chances of comparing the your method and the existing ones on some real-world benchmarks? For instance, [2, 3] target specifically on table understanding tasks, and CoT [4] and many others [1] are shown to be effective on various general language / math / code tasks.

- The authors only conduct their experiments based on the GPT-3.5 model, which is less satisfying. How does the base model selection affect the general performance? I think testing the method on other base models would convince readers that the method itself is generalizable in terms of LLMs. Also, studying your method based on some open-source LLMs would potentially shed more insights.

----
### References
[1] Chen, Wenhu, et al. "Program of thoughts prompting: Disentangling computation from reasoning for numerical reasoning tasks." arXiv preprint arXiv:2211.12588 (2022).

[2] Sun, Zhenjie, et al. "Table as Thought: Exploring Structured Thoughts in LLM Reasoning." arXiv preprint arXiv:2501.02152 (2025).

[3] Wang, Zilong, et al. "Chain-of-table: Evolving tables in the reasoning chain for table understanding." arXiv preprint arXiv:2401.04398 (2024).

[4] Wei, Jason, et al. "Chain-of-thought prompting elicits reasoning in large language models." Advances in neural information processing systems 35 (2022): 24824-24837.

**Questions:**

See weakness.

---

### Official Review · Reviewer_Mp4D · 2025-10-30

**Soundness:** 2
**Presentation:** 2
**Contribution:** 1
**Rating:** 2
**Confidence:** 4

**Summary:**

This paper proposes a new prompting framework for large language models that aims to overcome the scalability limitations of existing reasoning methods like Chain of Thought and tot.

**Strengths:**

The proposal of LWT as a natural language abstraction for defining step dependencies is conceptually elegant and aligns with how LLMs process structured input. The experiments are comprehensive, comparing XNoT against a wide range of strong baselines across multiple reasoning domains. The authors provide both empirical results and theoretical analysis to justify the benefits of modular decomposition, offering insights into why the method improves stability and performance. The reported gains in accuracy for larger inputs and significant reduction in prompt engineering effort are convincing and demonstrate practical value.

**Weaknesses:**

First, the core novelty—LWT—is mainly described in text without sufficient concrete examples or visualizations to clarify how real tasks are represented or executed. The reader must infer how the format translates into actionable steps, making reproducibility difficult.

2. the “intelligence amplification” claim is overstated; the method still depends heavily on predefined templates and task examples, which contradicts the claim of full automation. The process of generating and executing LWT scripts still involves manual tuning (e.g., constant prompts, example crafting), and the paper does not quantify how much of the design is truly automated. Third, the experiments rely mainly on synthetic or simple tasks (sorting, keyword counting, arithmetic), which do not strongly demonstrate general reasoning or transferability to realistic domains. There is no evidence that XNoT scales to tasks requiring long-context reasoning, multimodal input, or ambiguous natural language understanding. The comparison to baselines may also be unfair—since ToT and GoT require external scripts, their implementations here may have been simplified, which could bias the cost and accuracy metrics



 the theoretical section (Theorem 1, Lemma 1–2) is mathematically very shallow and does not connect to observed empirical patterns beyond verbal justification!!! Seems like just layering of formulas over simple methods.

**Questions:**

see above

---

### Official Review · Reviewer_mTVa · 2025-10-31

**Soundness:** 2
**Presentation:** 3
**Contribution:** 3
**Rating:** 4
**Confidence:** 4

**Summary:**

This paper proposes Executable Network of Thoughts (XNoT), a prompting framework where LLMs automatically generate and execute structured reasoning workflows using a LLM Workflow Template (LWT). By representing task decomposition and dependencies explicitly in natural language, XNoT achieves better scalability and cost efficiency on symbolic, arithmetic, and language reasoning benchmarks.

**Strengths:**

1. This work introduces an automatic, self-generated workflow framework that reduces human prompt engineering.

2. The results demonstrate that decomposing reasoning into multiple smaller LLM calls helps generalization and robustness under larger input scales.

3. The paper provides extensive task illustrations and detailed analyses that help understanding.

**Weaknesses:**

1. Despite the term network of thoughts, the LWT workflow actually executes linearly without backtracking or re-planning, which cannot recover from intermediate flaws and leads to error accumulation issues.

2. The generalizability of the workflow plan is unclear. Since the plan is generated once and remains static during execution, the model cannot adaptively update or leverage intermediate results. This design may work well for structured tasks with predictable execution steps but seems less suitable for more challenging unstructured reasoning scenarios.

3. Task decomposition itself is not new. XNoT mainly automates this process and executes the resulting plan sequentially. While the LWT format provides a cleaner structure, the overall design does not fundamentally address the core compositionality challenge in generalization.

**Questions:**

1. The execution workflow appears static once generated. How is the accuracy or reliability of the plan ensured, and can the workflow adapt or revise itself during execution?

2. Minor: There is no need to include percentages for every cell in Table 2

3. Typo: Line 396, nea -> near

---

### Official Review · Reviewer_GCGj · 2025-11-01

**Soundness:** 1
**Presentation:** 2
**Contribution:** 1
**Rating:** 2
**Confidence:** 5

**Summary:**

This paper proposes a prompting method that enables LLMs to generate and execute their own solution plans. The authors evaluate the approach on GPT-3.5-turbo and multiple synthetic datasets, demonstrating its superiority over prior methods such as CoT and ToT.

**Strengths:**

This paper endeavors to provide a clear description and theoretical analysis of the proposed prompting strategy and conducts extensive comparisons with prior prompting methods.

**Weaknesses:**

1. Only one model (GPT-3.5-turbo) is evaluated to demonstrate the efficacy of the proposed prompting method, and this model is somewhat outdated.
2. The evaluated datasets are either synthetic or relatively simple (e.g., GSM8K).

Given these limitations, it is unclear whether the proposed prompting method remains valuable in the current landscape, where reasoning models and more complex math or agent tasks are available. While prompting engineering is still an important direction, the experiments in this paper do not convincingly demonstrate its effectiveness and necessity.

**Questions:**

Why was GPT-3.5-turbo chosen as the evaluated model?

---

### Note · Authors · 2025-12-03

I have read and agree with the venue's withdrawal policy on behalf of myself and my co-authors.